# A Foundation Model for Patient Behavior Monitoring and Suicide Detection

## Abstract

Foundation models have achieved remarkable success across various domains, yet their adoption in healthcare remains limited, particularly in areas requiring the analysis of smaller and more complex datasets. While foundation models have made significant advances in medical imaging, genetic biomarkers, and time series from electronic health records, the potential for patient behavior monitoring through wearable devices remains underexplored. Wearable device datasets are inherently heterogeneous and multisource and often exhibit high rates of missing data, presenting unique challenges. Notably, missing patterns in these datasets are frequently not-at-random, and when adequately modeled, these patterns can reveal crucial insights into patient behavior. This paper introduces a novel foundation model based on a modified vector quantized variational autoencoder (VQ-VAE), specifically designed to process real-world data from wearable devices. Our model excels at reconstructing heterogeneous multisource time-series data and effectively models missing data patterns. We demonstrate that our pretrained model, trained on a broad cohort of psychiatric patients with diverse mental health issues, can perform downstream tasks without fine-tuning on a held-out cohort of suicidal patients. This is illustrated through the use of a change-point detection algorithm that identifies suicide attempts with high accuracy, matching or surpassing patient-specific methods, thereby highlighting the potential of VQ-VAE as a versatile tool for behavioral analysis in healthcare.

## 1 Introduction

The advent of foundation models (FMs) has catalyzed transformative advancements across various domains, from natural language processing to computer vision, achieving remarkable generalization across diverse tasks (Bommasani et al., 2021). However, their integration into healthcare has been comparatively slower. This delay can be attributed to clinical data's inherent complexity and variability and the challenges posed by heterogeneous, high-dimensional, and often incomplete datasets, such as electronic health records (EHR) (Moor et al., 2023).

An underexplored but crucial area in healthcare is the analysis of time-series data from wearable devices, which are increasingly used in daily life and provide a vast amount of data. This data presents several challenges: it is multisource (e.g., heart rate, motion, sleep patterns), heterogeneous (coming from different sensors with varying formats), and often incomplete, with significant portions missing due to device issues or user behavior (Wu et al., 2022; Lin et al., 2020). Importantly, these missing data points might hold valuable insights into patient behavior, so properly modeling them is crucial. An emerging field within computational psychiatry leverages data from wearable devices for early detection and personalized treatment of mental health conditions. By analyzing the continuous stream of data from sources such as heart rate variability and sleep patterns, researchers can detect behavioral changes that may indicate the onset or worsening of psychiatric, and more broadly, brain disorders (Wang et al., 2016; Thieme et al., 2020; Chekroud et al., 2021; Büscher et al., 2024).

To fully harness the potential of this data, models must handle the complexity of multisource, heterogeneous samples and account for missing information. Also, models should capture meaningful patterns from the missing data, as missingness often carries significant details on patient behavior. For instance, a wearable device that stops collecting data intermittently during certain times may indicate behavioral patterns such as sleep disturbances or irregular daily routines relevant to mental

health monitoring. Current state-of-the-art FMs, while powerful, struggle to handle this complexity or fully extract the valuable information embedded within such datasets.

Much effort has been focused on tasks such as data imputation, synthetic data generation, and anomaly detection within the broader field of deep generative models. Generative adversarial networks (GANs) have set the standard for high-resolution image generation, synthetic data creation, and domain adaptation. However, GANs do not provide latent-space encoders and are prone to mode collapse (where the model generates limited output diversity) (Grover et al., 2018). Alternatively, despite their success as the backbone of FMs in language and vision, transformers autoregressive models and diffusion models face obstacles in healthcare (Denecke et al., 2024; Xie et al., 2022). Their high computational cost, less interpretable continuous and hierarchical representations, and need for large datasets make them less ideal in domains like healthcare, where data is often scarce or expensive to collect (Wornow et al., 2023).

Variational autoencoders (VAEs) offer structured latent representations that enable data reconstruction and generation while explicitly modeling uncertainties. Additionally, VAEs naturally handle missing data by modeling the distribution of the underlying data, allowing them to fill in gaps and predict missing entries with a probabilistic approach, essential in healthcare applications involving incomplete and heterogeneous datasets (Collier et al., 2021). However, their extension to temporal settings is not trivial (Lucas et al., 2019), and they face optimization issues (e.g., posterior collapse, (Girin et al., 2022)) while employing continuous, rather than discrete, representations. Discrete representations improve interpretability and capture distinct patterns, particularly useful in applications where human understanding of the model is critical. As we will show in this work, this can be achieved with the so-called vector quantized-variational autoencoder (VQ-VAE) (van den Oord et al., 2018). VQ-VAE uses vector quantization and nearest-neighbor lookup to map features into discrete latent vectors, which store relevant information and capture complex relationships in the data. This is especially advantageous in cases where discrete states (e.g., different health states or behaviors) need to be represented.

In this work, we demonstrate how FMs constructed using VQ-VAEs can be leveraged to handle missing data in complex temporal databases, focusing on wearable device datasets. These FMs facilitate data reconstruction and subsequent downstream tasks, such as effective change point detection methods, underscoring the broader implications for personalized healthcare monitoring. Our contributions are twofold:

- We present a new foundation model built to process real-world data from various wearable devices and smartphones. This model is based on an enhanced version of the VQ-VAE, which is pretrained to reconstruct multisource, heterogeneous time-series data, model missing entries, and capture the underlying patterns of missingness.

- We demonstrate the versatility of our pre-trained model by using its internal discrete latent codebook to perform downstream medical tasks for which the model was not specifically trained. We highlight that no fine-tuning is required to achieve our results. Specifically, we develop a probabilistic change-point detection (CPD) algorithm for suicide detection that leverages the foundation model in an unsupervised manner. In particular, our model uses the encoded discrete latent codeword associated with the patient sequences generated by the VQ-VAE as input to the CPD algorithm. We show that this algorithm achieves an area under the curve of 0.92 when trying to predict events of suicidal nature based on the patient's behavior. We compare this value with a baseline patient-specific profiling method based on mixture models (AUC of 0.93) which requires an independent model trained per every patient in the dataset. Conversely, our VQ-VAE choice handles the generation of profile representations for all patients in the cohort at once in a unique model, thereby achieving higher computational efficiency and facilitating scalability.

## 2 BEHAVIORAL DATASET

The widespread use of personal digital devices, such as smartphones and wearables, has enabled the passive collection of behavioral metrics, such as the pattern of mobile apps used, distance traveled, time spent at home, and sleep patterns. This method, known as passive digital phenotyping (PDP), allows for continuous, unobtrusive monitoring without requiring active user input, making

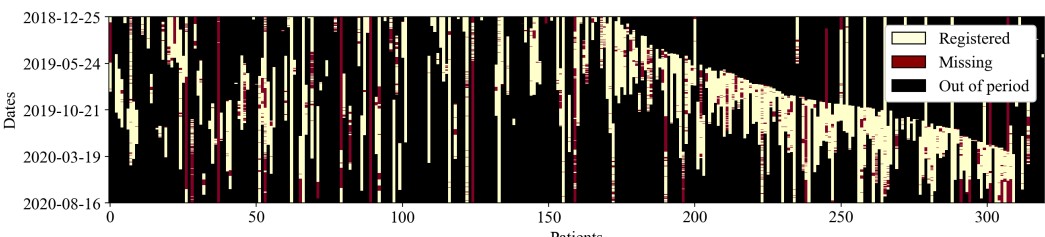

Figure 1: Visualization of data missingness. The availability of step count data is displayed over approximately one-and-a-half years. The length of registered periods varies from patient to patient, and most contain scattered days or sequences with no data.

it ideal for long-term monitoring. These data streams have proven valuable for characterizing and tracking psychiatric patients (Moreno-Muñoz et al., 2020; Romero-Medrano & Artés-Rodríguez, 2023; Büscher et al., 2024). Recent research has applied PDP to detect behavioral shifts that may indicate serious mental health risks. For instance, the SmartCrisis study (Berrouiguet et al., 2019) developed a personalized suicide prevention strategy by monitoring participants with a history of suicidal behavior over extended periods.

A common challenge in PDP studies is missing data, often caused by smartphone operating systems terminating background processes or patients intentionally discontinuing the use of their wearable devices. These disruptions, essential for passive data collection, result in significant gaps in the data stream, compromising the quality and completeness of the dataset (see Figure 1 for a representative example). Additionally, the collected data are heterogeneous: some variables are recorded as daily summaries with limited dimensions (e.g., sleep duration, start and end times), while others provide more granular, time-segmented information, such as physical activity or app usage time.

The dataset used in this work was collected via a PDP-enabled mobile application provided by *Company A* and serves as the basis for model training, validation, and testing.[1] It contains 1,122,233 entries across 64 variables, comprising data from 5,532 patients enrolled in 39 clinical programs. The collection period spans from January 1, 2016, to March 13, 2024. Each entry encapsulates aggregated daily metrics from original time-stamped recordings captured at 30-minute intervals across multiple sensors. One of the main challenges this dataset presents is the high proportion of missing data, particularly for variables where data collection was frequently interrupted. To address this, we focused on a subset of variables with a missingness rate below 85%. Table 1 overviews the selected variables, their types, and the corresponding missingness rates. The dataset also contains significant noise and outliers, likely due to sensor malfunctions, inconsistent user behavior, environmental factors, and hardware or software issues. A detailed description of the dataset and its preprocessing is provided in Appendix A.

## 3 VQ-VAE AS A FOUNDATION MODEL

The vector quantized-variational autoencoder (van den Oord et al., 2018) extends the traditional VAE by incorporating a discrete latent space, addressing some of the limitations of continuous representations. In VQ-VAE, the latent space is composed of $K$ discrete embeddings, $\mathbf{e}_j \in \mathbb{R}^D$, where $j \in \{1, 2, \ldots, K\}$, forming the codebook $E = \{\mathbf{e}_j\}_{j=1}^K$. The encoder produces a continuous latent output $\mathbf{z}_e(\mathbf{x})$, which is quantized to the nearest embedding $\mathbf{e}_k$ using nearest-neighbor lookup:

$$q(z = k | \mathbf{x}) = \begin{cases} 1 & \text{for } k = \arg\min_j \|\mathbf{z}_e(x) - \mathbf{e}_j\|_2, \\ 0 & \text{otherwise} \end{cases} \tag{1}$$

where $z \in \{1, \ldots, K\}$ indicates that $\mathbf{z}_q(\mathbf{x}) = \mathbf{e}_k$ from the codebook $E$ to which we map the enconder output $\mathbf{z}_e(\mathbf{x})$. Hence, $\mathbf{z}_q(\mathbf{x})$ denotes the decoder input. The loss function takes the following form

$$L = \underbrace{\log p(\mathbf{x}|\mathbf{z}_q(\mathbf{x}))}_{\text{Reconstruction loss}} + \underbrace{\|\text{sg}[\mathbf{z}_e(\mathbf{x})] - \mathbf{e}_k\|_2^2}_{\text{Codebook loss}} + \beta \underbrace{\|\mathbf{z}_e(\mathbf{x}) - \text{sg}[\mathbf{e}_k]\|_2^2}_{\text{Commitment loss}}, \tag{2}$$

---

[1]The company name has been anonymized for the review process.

Table 1: Type and relative missingness of selected variables.

| Category | Variable name | Type | Relative missingness (%) |
|---|---|---|---|
| **Activity** | Time Walking (s) | $\mathbb{R}_{\geq 0}$ | 62.79 |
| | App Usage Total (s) | $\mathbb{R}_{\geq 0}$ | 83.15 |
| | Practiced Sport[3] | $\{0, 1\}$ | 0.00 |
| | Total Steps | $\mathbb{N}_0$ | 55.30 |
| **Location** | Location Clusters Count[4] | $\mathbb{N}_0$ | 72.53 |
| | Traveled Distance (m) | $\mathbb{R}_{\geq 0}$ | 73.01 |
| | Time at Home (m) | $\mathbb{R}_{\geq 0}$ | 82.53 |
| **Other** | Weekend[5] | $\{0, 1\}$ | 0.00 |
| **Sleep** | Sleep Duration (s) | $\mathbb{R}_{\geq 0}$ | 66.76 |
| | Sleep Start (s)[6] | $\mathbb{R}$ | 66.11 |

where $\text{sg}[\cdot]$ denotes the stop-gradient operator. The reconstruction loss is optimized by both the encoder and decoder, forcing them to provide relevant data representations. The codebook loss ensures that the embeddings capture such representations. The commitment loss enforces stability during training by limiting the updates in encoder output to match current embeddings.[2]

## 3.1 MODELING MISSING DATA

A key challenge in real-world healthcare datasets, especially time-series data from wearable devices, is missing data. We handle missing data by extending the VQ-VAE architecture to jointly model both the observed data and the missingness pattern. Let $\mathbf{x}_d^{(i)} \in \mathbb{R}^T$ represent the real-valued time-series data vector of length $T$ for patient $i$ and variable $d$, where each component corresponds to a data entry at a sampled time instant and $d \in \{1, \ldots, D\}$. Recall that the set of possible variables are summarized in Table 1. Let $\mathbf{m}_d^{(i)} \in \{0, 1\}^T$ denote a binary mask vector where each entry indicates whether the corresponding entry is observed (entry value equal to 1) or missing (entry value equal to 0). The corrupted signal, after applying the binary mask $\mathbf{m}_d^{(i)}$, is defined as:

$$\tilde{\mathbf{x}}_d^{(i)} = \mathbf{m}_d^{(i)} \odot \mathbf{x}_d^{(i)}, \tag{3}$$

where $\odot$ denotes the element-wise (Hadamard) product. This formulation applies zero-imputation, ensuring missing data points do not introduce misleading information, as gradients related to imputed values remain zero during backpropagation (Nazábal et al., 2020).

Inspired by (Collier et al., 2021) for VAEs, we propose three VQ-VAE variants (see Figure 2) that incorporate the missing mask within the VQ-VAE structure: Model A0: No missingness mask conditioning; ii) Model A1: Missingness mask conditioning in the encoder only; iii) Model A2: Missingness mask conditioning in both encoder and decoder. Model A0 follows a simpler architecture, where only the input signal is processed, without incorporating any missingness mask in either the encoder or decoder stages. As a result, model A0 relies solely on the zero-imputed signal.

In models A1 and A2, both the input signal and missingness mask are integrated within the encoder. The missingness mask is pre-processed through $M$ convolutional layers, which allow the model to capture dependencies in the missing data patterns across variables. The processed mask is concatenated with the input signal along the channel axis, and the combined data is passed through $N$ convolutional layers, resulting in a continuous latent representation. This latent representation is then quantized via a nearest-neighbor lookup in the codebook before being passed to the decoder.

---

[2]As described in van den Oord et al. (2018), the codebook loss can be replaced by exponential moving averages (EMA) of $\mathbf{z}_e(x)$, which is the implementation used for the experiments in this work

[3]Sports activity is flagged if the combined time spent walking, running, bicycling, and other sports exceeds one hour.

[4]Locations are dynamically defined by clustering algorithms grouping related geographical positions.

[5]1 represents weekend data, while 0 represents weekday data.

[6]The reference time is 23:00. Negative values indicate seconds before this time, and positive values indicate seconds after.

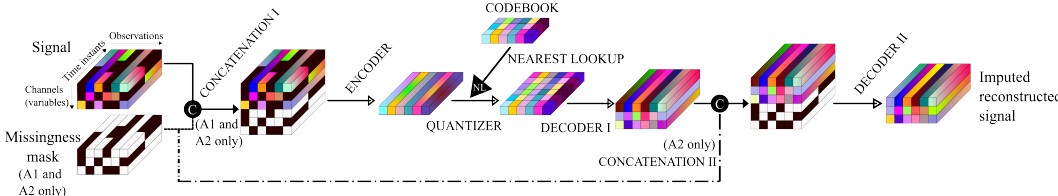

(a) Overview of the variant VQ-VAE structure. The complete set corresponds to model A2. Model A1 only features encoder conditioning and model A0 does not present any missingness mask concatenations, operating solely on the signal.

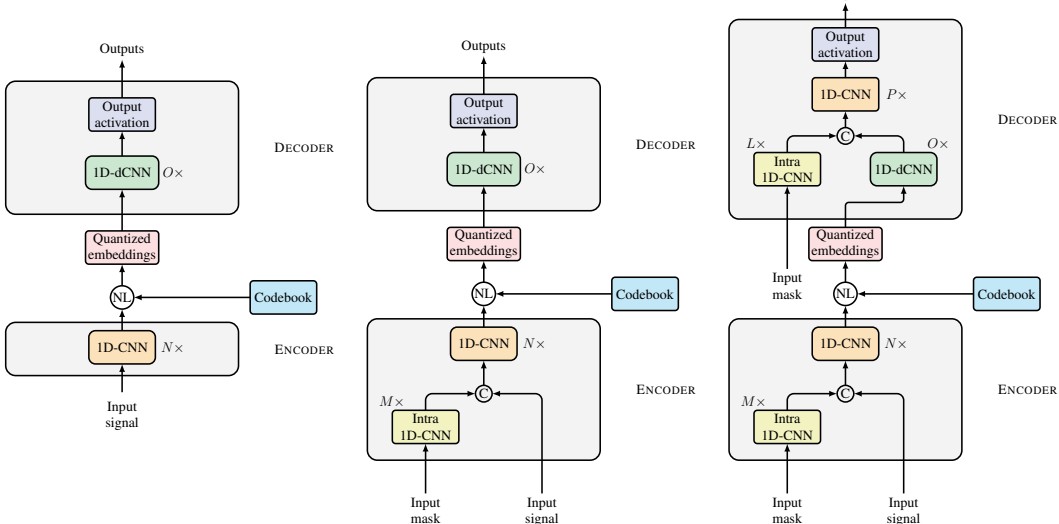

(b) Model A0 (without missingness mask conditioning).

(c) Model A1 (encoder-only missingness mask conditioning).

(d) Model A2 (encoder-decoder missingness mask conditioning).

Figure 2: Overview of proposed missing-aware VQ-VAE variants.

In model A1, the quantized embeddings are further processed through $O$ deconvolutional layers, followed by variable-specific activation functions tailored to the data type. In contrast, model A2 employs a more complex structure: the quantized embeddings are concatenated with the separately processed missingness mask (which is transformed via $L$ convolutional layers) along the channel axis before passing through additional $P$ convolutional layers. The output is then fed into variable-specific activation functions.

Using the proposed variant VQ-VAE architectures, we trained the model on the PDP behavioral dataset described in Section 2. Each data modality was modeled by selecting an appropriate likelihood function tailored to its distributional characteristics. For real-valued variables, we employed a Gaussian likelihood, while for binary features, a Bernouilli likelihood was used. Count data were presented over a sufficiently extended array of values, and the Gaussian likelihood was also applied to them. For more information on data preprocessing, see Appendix A.

Models were trained according to their reconstruction performance on observed data, and they were analyzed on their ability to impute artificially-introduced missing data (see Section 5). Detailed architecture specifications are provided in Appendix B of the supplementary material.

## 4 CHANGE-POINT DETECTION

CPD involves identifying abrupt shifts in a time series. The objective is to segment sequential data into partitions generated under different underlying conditions, without prior knowledge of when these changes occur (Page, 1955). The mathematics behind this model are developed in this section, followed by an explanation of how CPD can be integrated as a downstream task of the VQ-VAE.

## 4.1 BAYESIAN ONLINE CPD

A Bayesian online approach, presented in Adams & MacKay (2007), confronts the CPD problem from a probabilistic perspective. This framework assumes that the observed data at day $t$—or the latent profiles constructed from them— are generated by some mathematical distribution with unknown parameters $\theta_t$. Each assumed partition is independent of the others and defined by unique parameters. At the same time, observations are regarded as samples drawn from those partitions in an independent and identically distributed (i.i.d.) manner. A significant shift in the base parameters of the distribution will be considered a change point. In the following, subscripts refer to a specific element or sequence from temporal variables. For example, the term $\mathbf{z}_t$ refers to the $t$-th element of the corresponding sequence, while $\mathbf{z}_{1:t}$ indicates the span from the first observed day until the current date $t$.

We introduce the counting variable $r_t \in \mathbb{N}_0$ to denote the *run length* at day $t$, representing the time (in units, e.g., days in our setting) that elapsed since the last change point. For a given day $t$, the run length can either increase by one if no change is detected or drop to zero otherwise. Hence, our model focuses on inferring the posterior distribution of this variable, given by

$$p(r_t|\mathbf{z}_{1:t}) = \frac{p(r_t, \mathbf{z}_{1:t})}{p(\mathbf{z}_{1:t})}, \tag{4}$$

which can be made in a recursive and online manner, meaning that, given all past observations, the probability that a change occurred is distributed along all previous days. By deriving this run length distribution for every day, we can have a sense of how our signal behaves in time and when a substantial change has occurred. The run length $r_t$ and the observed data (patient profiles in our work) $\mathbf{z}_t$ are jointly modeled as

$$p(r_t, \mathbf{z}_{1:t}) = \int p(r_t, \mathbf{z}_{1:t}, \theta_t) \, \mathrm{d}\theta_t, \tag{5}$$

where the model parameters are marginalized. The joint density within the integral can be factorized by marginalizing over the run length of the previous day, $r_{t-1}$, which we assume has been previously obtained, as follows:

$$p(r_t, \mathbf{z}_{1:t}, \theta_t) = \sum_{r_{t-1}} p(r_t, r_{t-1}, \mathbf{z}_{1:t}, \theta_t) \tag{6}$$

$$= \sum_{r_{t-1}} \underbrace{p(r_t|r_{t-1})}_{\text{change point prior}} \underbrace{p(\mathbf{z}_t|\theta_t)p(\theta_t|r_{t-1}, \mathbf{z}_{1:t-1})}_{\text{predictive posterior}} \underbrace{p(r_{t-1}, \mathbf{z}_{1:t-1})}_{\text{recursive term}}. \tag{7}$$

The prior probability of having a change point at any moment, conditioned on past change-points, is defined by the hazard function $H(\cdot)$ (Ibe, 2014), which in our case was set to a constant that depends on some hyperparameter $\lambda$ such that $p(r_t|r_{t-1}) = H(r_{t-1}) = 1/\lambda$. The recursive term in Equation 6 is independent of the model parameters and can be computed recursively. Thus, it follows that

$$p(r_t, \mathbf{z}_{1:t}) = \sum_{r_{t-1}} p(r_t|r_{t-1})\pi_t p(r_{t-1}, \mathbf{z}_{1:t-1}), \tag{8}$$

where the term $\pi_t$ denotes the predictive posterior of the next datum conditioned to past run length and observed data, which is given by

$$\pi_t = p(\mathbf{z}_t|r_{t-1}, \mathbf{z}_{1:t-1}) = \int p(\mathbf{z}_t|\theta_t)p(\theta_t|r_{t-1}, \mathbf{z}_{1:t-1}) \, \mathrm{d}\theta_t. \tag{9}$$

The complexity of this term is determined by the choice of prior and likelihood distributions that define the data. In fact, its computation is often intractable, unless the underlying process is modeled after an exponential family with conjugate prior (Turner et al., 2013). However, other strategies can be employed to obtain an approximation of the predictive posterior, such as Markov chain Monte Carlo methods (Moreno-Muñoz et al., 2019). In our case, we exploit the simplicity of the VQ-VAE patient encoding, as it yields a sequence of categorical observations, to implement a robust CPD with inference in closed-form expression.

Once all probabilities are derived, Equation 4 returns the run length characterization of the complete temporal sequence: for each day, a distribution explains how the probability of a potential change

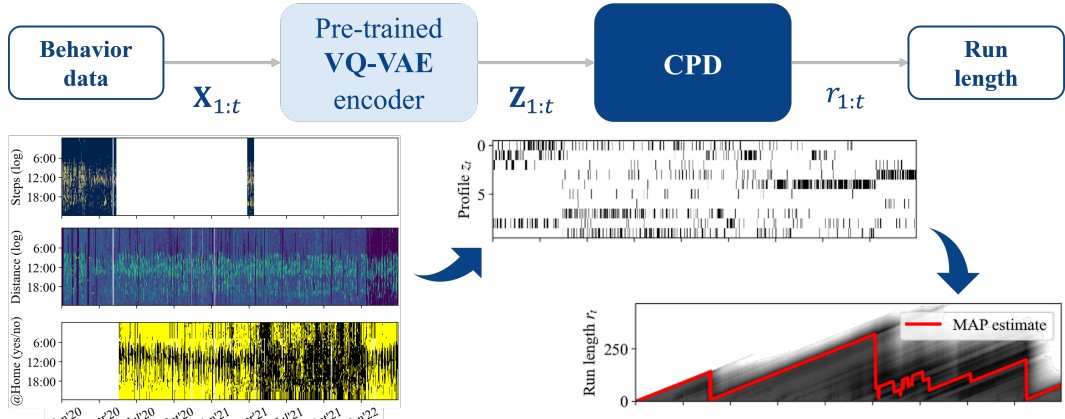

Figure 3: Diagram of the VQ-VAE–CPD integration, including the mathematical notation for each variable at each step: observed data ($\mathbf{X}_{1:t}$), discrete latent profiles ($\mathbf{Z}_{1:t}$), and run length prediction ($r_{1:t}$). Boldfaced, capitalized notation denotes the concatenation of data examples and their respective latent representations. The plots below the diagram illustrate a real-world example: three behavioral sources (step count, distance traveled, and time spent at home) are compressed into a latent profile, which is then used to compute the run length, i.e., the time since the last change point. The red line shows its MAP estimation (the most probable run length for each day).

point is shared among all previous days. Subsequently, a *maximum a posteriori* (MAP) estimation is performed to identify the most likely run length for every day. The CPD output is a binary prediction vector, where 1 indicates a detected change point and 0 otherwise. Various methods involving a decision threshold can be employed to process the MAP estimation into this binary variable, which is necessary to contrast model predictions against real events. Please refer to Appendix D for a more in-depth description of the CPD algorithm.

## 4.2 CPD AS A DOWNSTREAM TASK

Online CPD has demonstrated promising results in real-world applications, such as water quality monitoring (Ba & McKenna, 2014) and the analysis of epileptic activity (Malladi et al., 2013). However, its application to human behavior analysis is just commencing to be explored. This context often involves high-dimensional, heterogeneous, periodic variables with a significant rate of missing entries (Reinertsen & Clifford, 2018; Bloom et al., 2024), characteristics that impose some unique challenges in their analysis. Specifically, the high dimensionality of the dataset described in Section 2 can complicate the estimation of underlying parameters and the posterior probability of the run length. Past work has employed heterogeneous mixture models (HetMM) to address this issue as a profiling step prior to the CPD stage (Moreno-Muñoz et al., 2019). Similar to the VQ-VAE, HetMM assume that the observed high-dimensional data can be generated from a latent, lower-dimensional variable, allowing to represent each time point with a characteristic profile. The CPD model can then analyze the pattern of these profiles over time to identify changes in behavior.

HetMM methods have proven effective in integrating variables of diverse statistical types and handling partially missing data, especially for suicide prediction (Moreno-Muñoz et al., 2020). However, these approaches lack scalability and efficiency, as each individual is represented by a separate model trained on their own data. While this allows for personalized modeling, it necessitates an independent model per user, increasing computational requirements and hindering the ability to identify shared patterns across individuals. Although this may not be problematic for small datasets, it becomes a major limitation in large-scale applications or real-time analysis, where computational efficiency is essential.

The VQ-VAE foundation model proposed in this paper offers a compelling alternative to overcome these limitations. The VQ-VAE encoder's discrete latent representations serve as lower-dimensional profiles, analogous to those produced by HetMM, and can be used as inputs to the CPD model for change-point detection. To evaluate this integration, we tested it on a held-out cohort not involved in VQ-VAE training. These patients, part of a suicide prevention program, had behavior data collected through passive digital phenotyping and clinical records of suicide attempts or emergency visits due

to self-harm. As deviations in daily routines often precede such crisis events, this cohort provides a strong basis to validate CPD accuracy. Figure 3 summarizes the complete pipeline.

One of the main advantages of the foundation model is that, unlike the HetMM case, a single VQ-VAE model is trained over a broader population to produce latent profiles. This paradigm shift supposes an improvement in efficiency and scalability: an increase in the number of individuals does not imply defining and storing more models, each with a new set of parameters to be tuned, but instead leads to the very same model being trained on a larger dataset (i.e., during more epochs). Moreover, this approach allows to jointly model behavioral data from various users across several cohort studies, capturing a richer perspective of human behavior. Still, perhaps the most compelling aspect of our proposed solution is that no fine-tuning is necessary on the pre-trained VQ-VAE to produce the patient profiles for CPD. Its success in solving the CPD task is an example of how the VQ-VAE foundational model presented in this work can be leveraged to potentially aid in the broader variety of health-related problems, as detailed in Section 3.

## 5 RESULTS

### 5.1 SELF-SUPERVISION THROUGH RECONSTRUCTION AND IMPUTATION

We evaluated three variants of our VQ-VAE model—A0, A1, and A2—on the PDP dataset described in Section 2. These models were trained using a similar objective to the original VQ-VAE in Equation 2, which includes reconstruction loss and commitment loss. However, instead of optimizing the codebook loss directly, we updated the codebook using exponential moving averages (EMA), as outlined in Section 3.

The models were trained specifically to reconstruct observed data, focusing on minimizing the reconstruction error for known data points. This approach prioritizes the quality of reconstructing available data without explicitly optimizing for imputing missing values. Consequently, evaluating their performance on data imputation under various missingness mechanisms provides a more rigorous test of their generalization capabilities in handling unobserved data, which they were not directly trained to predict.

We assessed the models' performance on both reconstruction and imputation tasks, which are crucial for evaluating their effectiveness in scenarios involving both observed and unobserved data. Reconstruction refers to recovering known values based on latent representations, while imputation involves estimating values that were not observed during training. For the imputation task, the models were exposed to synthetic missingness, simulating both missing completely at random (MCAR) and missing not at random (MNAR) mechanisms. In the MCAR setting, missing instances were introduced uniformly at random, whereas in the MNAR scenario, missingness was conditioned on the values of the target variables. This setup provides a comprehensive evaluation of the models' capabilities in both random and structured missingness settings.

Figure 4 presents a selection of representative signal reconstructions for both observed and imputed instances. These visualizations highlight the variant VQ-VAE models' ability to accurately recover data. Additional signal reconstructions and tables showing results on reconstruction and imputation quality, are provided in Appendix E.1 due to space constraints. Furthermore, our results show that the codebook usage per sample is usually very sparse for most patients, as can be checked in Appendix E.2.

### 5.2 SUICIDE DETECTION (DOWNSTREAM TASK)

The practical validity of the VQ-VAE model was assessed by integrating it with a CPD architecture to predict risk events in the context of suicide prevention, as explained in Section 4.2. The performance of the CPD coupled to the HetMM profiling stage was used as a benchmark for comparison.

When the run length estimation is transformed into a prediction sequence, a hyperparameter is involved to set the decision threshold for marking positives, i.e. crisis events. This threshold was swept to produce a receiver-operating characteristic (ROC) curve, which we used to assess the model trade-off between sensitivity (ability to correctly identify crisis events) and specificity (ability to not raise

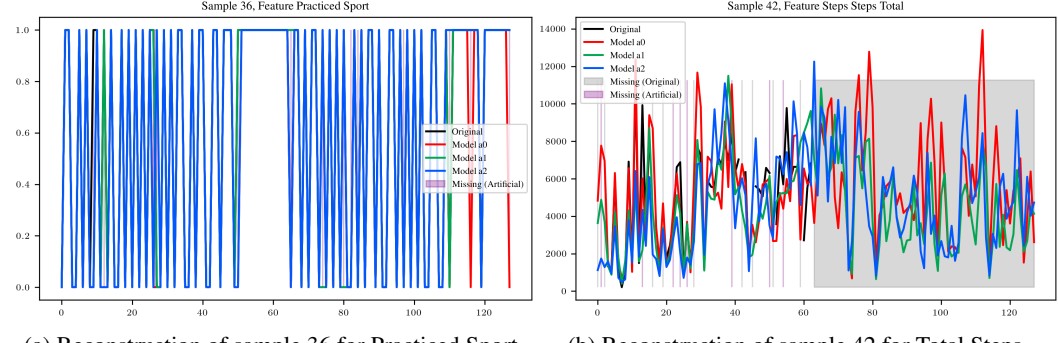

(a) Reconstruction of sample 36 for Practiced Sport.

(b) Reconstruction of sample 42 for Total Steps.

(c) Reconstruction of sample 180 for Time Walking.

(d) Reconstruction of sample 493 for App Usage Total.

Figure 4: Representative signal reconstructions for observed and imputed instances. In cases where the original signal is not explicitly shown, it is because one or more of the models (whose reconstructions are plotted) overlap the true signal precisely, obscuring the original data. Additional signal reconstructions are available in Appendix E.1.

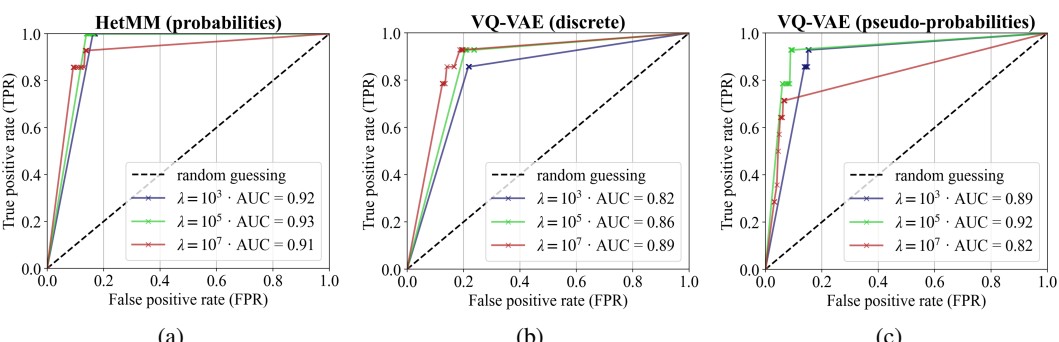

(a)

(b)

(c)

Figure 5: ROC curves comparing the performance of the CPD with three different versions of the prior profiling stage: (a) a heterogeneous mixture model, (b) our VQ-VAE using its discrete latent variable, and (c) the same VQ-VAE but returning the profiles as pseudo-probabilities. The three colored lines in each plot correspond to three different values of hyperparameter $\lambda$. The number of possible profiles ($K$) was set to 10 in the HetMM and 20 in the VQ-VAE. Version A0 of the VQ-VAE was used. AUC values are given in each plot.

false alarms, i.e., not returning a positive when there are no events). These metrics, together with the commonly used area under the curve (AUC), were used to compare the different model outputs.

Figure 5 compares the CPD performance using HetMM and VQ-VAE as profiling stages. The CPD implementation accepts either discrete (integer labels for daily profiles) or probabilistic (profile probabilities for each day) sequences. While HetMM naturally returns probabilistic profiles, VQ-VAE provides discrete profiles, which can increase noise when the confidence is low (i.e., the profile distribution is flat). To address this, we compute pseudo-probabilities for VQ-VAE profiles by calculating the Euclidean distances between continuous encoder outputs and latent embeddings,

and then applying a softmax transformation to the inverse of these distances. This way, embeddings closer to the input have higher probabilities, providing a probabilistic interpretation of the discrete latent profiles. Figure 5 displays CPD results for HetMM (probabilistic), VQ-VAE (discrete), and VQ-VAE (pseudo-probabilistic) profiles.

The experiment was run for different values of hyperparameter $\lambda$, involved in the so-called hazard function that defines the prior probability of having a change point at any given time instant. The performance of the CPD is affected by this hyperparameter, which can be tuned to adapt its sensitivity. Higher $\lambda$ decreases the change-point prior, minimizing the rate of true positives. The values we used for $\lambda$ are $10^3$, $10^5$ and $10^7$, with none of them significantly outperforming the others.

The reference mixture model (Figure 5a) maintained a high sensitivity ($y$-axis of the plot), often detecting 100% of the suicide events used as validation. This target was not achieved by the two VQ-VAE proposals, whose maximum sensitivity was 92.8%. Regarding specificity—represented in the $x$-axis of the ROC space—, the VQ-VAE discrete profiles yielded higher rates of false positives than the HetMM, indicating a lower specificity. Remarkably, the use of the VQ-VAE with pseudo-probabilities achieves comparable performance to the HetMM approach, sometimes even outperforming it, especially for large values of $\lambda$. Some of the tested models display false positive rates as little as 0.07 (i.e., 7% of false alarms) while still maintaining their sensitivity close to 80%. The VQ-VAE model with the best AUC score was the one using pseudo-probabilities for the patient profiling with $\lambda = 10^5$, achieving an AUC score of 0.92, which competes with the HetMM versions.

We emphasize the significance of this result, as the VQ-VAE approach uses a single model to extract patient profiles that are then used as inputs for the CPD algorithm, establishing a novel and scalable approach for suicide detection.

# 6 CONCLUSION

In conclusion, this paper presents a significant advancement in applying foundation models to the analysis of heterogeneous, multisource time-series data collected from wearable devices in healthcare. By leveraging the modified VQ-VAE architecture, our model addresses key challenges such as high rates of missing data and the complex nature of multisource inputs. The model's capacity to reconstruct missing entries and capture critical behavioral patterns through discrete latent representations enhances interpretability, positioning it as a powerful tool for healthcare applications. Our results demonstrate that the model, even without patient-specific fine-tuning, performs remarkably well in tasks such as change-point detection, accurately identifying critical events like suicide attempts. This highlights its potential in monitoring patient behavior and supporting early interventions in healthcare.

Moreover, the pre-trained model's success in downstream tasks, such as clustering patients using encoded latent sequences, underscores its adaptability and utility beyond the scope of its initial training. The ability to generalize across datasets and extract meaningful insights from missing data offers a new paradigm for patient monitoring, where passive behavioral data from wearable devices can be fully utilized. This work not only broadens the scope of foundation models in healthcare but also opens new avenues for integrating wearable technology into personalized medicine, with the potential to enhance patient outcomes through more precise and actionable behavioral analysis.

Future work could explore coupling the VQ-VAE with autoregressive models such as PixelRNN or PixelCNN for more sophisticated generative tasks. These extensions would enable realistic synthetic data generation by sampling in the latent space, which is particularly relevant in healthcare for tasks like simulating patient trajectories or generating synthetic datasets for rare conditions. Such developments could further advance the model's capability in predicting long-term health outcomes and in generating high-fidelity synthetic data, which is crucial for augmenting limited real-world datasets, particularly in scenarios involving rare diseases or underrepresented populations.

## ETHICS STATEMENT

The clinical program on suicide prevention whose cohort was involved in our downstream task was approved by *Institution B* and carried out in compliance with the tenets of the Declaration of Helsinki. All patients gave written informed consent to participate after a complete description of the study and they were not compensated for their participation. Similar circumstances surround the remaining 38 programs whose subjects were involved in the VQ-VAE training phase (additional details can be provided if our research is accepted). Concerning data protection and confidentiality, each patient's identification was ensured by a username and password. The data gathered by the *Company A* app were anonymized if it were sensitive data, then translated into a unique data schema, and finally transmitted through a secure Wi-Fi network to *Company A*'s backend server where it were stored.

## REPRODUCIBILITY STATEMENT

Our study uses a proprietary dataset collected from wearable devices, as described in detail in Section 2 and Appendix A. For data collection and preprocessing steps, we provide a comprehensive explanation, including methodologies for handling missing data and generating input sequences. If this work is accepted, we will release the source code for our VQ-VAE model variants introduced in Section 3 and further detailed in Appendix B in a GitHub repository. This repository will include code for model training, reconstruction, and imputation, along with pretrained models to facilitate reproducibility. The profiling preparation process for the CPD algorithm, which uses the encoder and codebook of the VQ-VAE model, is outlined in Appendix C. The code implementing this procedure will also be made available in the same GitHub repository.

Regarding the CPD algorithm, the mathematical concept behind is briefly covered in Section 4 and further details on hyperparameters involved are provided in Appendix D. More in-depth explanations on its implementation and integration with the heterogeneous mixture model are offered in some of our past research, and code scripts may be shared upon request.

Our supplementary materials and appendices provide all necessary details to enable reproducibility, including data processing scripts, experimental configurations, and hyperparameters used throughout the paper.

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

# A  DATA PREPROCESSING FOR THE VQ-VAE

As outlined in Section 2, the original dataset comprises 64 variables, many of which exhibit high levels of missing data. This poses a significant challenge for standard deep learning techniques, which typically require large datasets to generalize effectively. Thus, an extensive data processing pipeline was necessary and is described in detail here.

In order to rigorously assess the performance of the three proposed models (A0, A1, and A2), we implemented a robust evaluation strategy based on an $n$-partition scheme of the original dataset. Each partition was systematically allocated for training, validation, and testing—along with reconstructed signal plots—across all models. Importantly, this design ensured that the data partitions were consistent across all models, precluding any leakage of patient data between partitions within a given $n$-partition configuration. This strict partitioning protocol enabled a fair comparison between the mask-conditioned architectures (A1, A2), and the non-conditioned baseline model (A0), ensuring identical experimental conditions across different, randomly sampled sections of the dataset.

A key challenge in modeling time-series data is the transformation of the tabular dataset into a format suitable for deep learning techniques. Specifically, we reshaped the data into observation batches with dimensions $[B, F, L]$, where $B$ denotes the batch size, $F$ the number of features, and $L$ the sequence length. The initial preprocessing step involved the removal of uninformative or redundant variables, coupled with a stringent constraint ensuring that patient records were not split across training, validation, and test within any $n$-partition. Instead, all data from a single patient were placed within the same partition to preserve temporal and contextual consistency.

Several variables were excluded from the analysis due to inconsistencies in missing data reporting. For instance, features such as the variables measuring the minimum/maximum/average heart rate used a placeholder value of $-1$ to indicate missing data, whereas other variables adhered to the standard Numpy convention of using NaN. Date-related variables also required normalization to a consistent format. Additionally, certain variables contained erroneous or outlier values, likely due to faulty sensors or other external factors, as discussed in Section 2. While it was not possible to completely eliminate all erroneous entries due to the absence of key contextual variables, we removed the majority of manifestly inaccurate data points. For example, the *Sleep Duration* variable is known to be device-dependent, with different vendors applying varying algorithms to detect sleep patterns. Similarly, the *Total Steps* variable can be influenced by non-step movements, such as hand gestures, while the *App Usage Total* variable is constrained by vendor-specific limitations. The *Location Clusters Count* variable, being derived from external algorithms that process raw geolocation data, also exhibited potential inaccuracies.

To mitigate these issues and improve model stability, we applied the constraints shown in Table 2, where the columns "Minimum Bound" and "Maximum Bound" specify the ranges to clip the values in "Original Minimum" and "Original Maximum". Any value outside these bounds was marked as missing.

Table 2: Clipping constraints applied to ensure model stability. The *Original Minimum* and *Original Maximum* columns represent the range of raw variable values in the dataset, while the *Minimum Bound* and *Maximum Bound* columns define the clipping thresholds. Values falling outside these bounds were treated as missing to avoid outliers, erroneous data, and ensure more reliable model training.

| Variable | Original Minimum | Original Maximum | Minimum Bound | Maximum Bound |
| --- | --- | --- | --- | --- |
| Sleep Start (s) | -11,657,590 | 7,430,400 | -22,500 | 25,000 |
| Traveled Distance (m) | 7.891e-10 | 9,945,435.20 | 20 | 95,000 |
| Time at Home (m) | 0.0 | 1,440 | 120 | — |
| Sleep Duration (s) | 1.0 | 86,400.0 | 3,600 | 54,000 |
| Time Walking (s) | 0.0 | 3,098,824.0 | 120 | 15,000 |
| App Usage Total (s) | 0.0 | 630,478.0 | 180 | 35,000 |
| Location Clusters Count | 0 | 40 | 1 | 15 |
| Total Steps | 1 | 99,734 | 150 | 25,000 |

After the initial preprocessing steps, we ensured that each patient's time-series data remained temporally contiguous. Specifically, if a patient's records spanned from March 15, 2019, to May 2, 2019, but included a gap until May 15, 2019, the data were split into two distinct sequences: one

from March 15 to May 2, and the other from May 15 to the end of the recording period (e.g., June 24). Sequences that were shorter than the predefined minimum length, were discarded to maintain consistency in sequence length across the dataset. This was not applied to the final subset of held-out psychiatric patients whose time-series—varying in length— were processed in full.

Next, we addressed differences in scale across continuous and counting variables by applying appropriate transformations. For real-valued continuous features, we utilized scikit-learn's `RobustScaler`, which is well-suited for handling data with outliers by centering the data around the median and scaling it based on the interquantile range (IQR). These transformations were fitted on the training set and subsequently applied to the validation and test sets to ensure consistency across all partitions.

It is important to note that all metrics and signal reconstructions reported in this work reflect the original feature space. To achieve this, we reversed the scaling transformations prior to computing evaluation metrics and generating signal plots. This approach ensures that the reported results are both interpretable and faithful to the original data distributions.

For each model instance, a missingness mask was dynamically generated for each patient sequence, with synthetic missingness introduced to simulate unobserved data. This missingness mask consisted of three distinct values: "0" for originally missing data, "1" for observed data, and "2" for synthetically induced missing data. However, for model input, the mask was binarized by collapsing "2" into "0", as the model was designed to treat all missing entries uniformly, regardless of whether the missingness was natural or synthetically generated.

To simulate missing data, we employ two distinct strategies: MCAR (missing completely at random) and MNAR (missing not at random). Each mode is constructed to introduce missingness in ways that reflect both random and structure data loss.

In the MCAR setting, missingness is introduced through a random process designed to target approximately 10% of the observed entries. However, a series of safeguard conditions modulate this target to ensure data integrity. Specifically:

- If more than 85% of the data for any feature is already missing, no additional missingness is introduced.
- A flat rate of 10% is tentatively introduced if there is not prior existing missingness for a given sample.
- For each feature, missing values are added by randomly selecting from the observed entries, ensuring that only those entries are affected.

The result is a systematic, yet random, distribution of missingness that prevents over-saturation while maintaining stochasticity.

In contrast, MNAR employs a feature-drive approach, introducing missingness based on relationships between variables and their values. Structured missingness is inserted through a combination of non-linear conditions and thresholds. The MNAR process unfolds as follows:

- If more than 85% of the data for any feature is already missing, no additional missingness is introduced.
- Non-linear conditions are applied to enforce missingness. For example, if a feature consistently deviates from its typical range (e.g., extreme values of a continuous variable), missingness is introduced.

To avoid excessive data sparsity, the same 85% ceiling on missingness per feature is applied, ensuring that no single features becomes overwhelmingly absent. Furthermore, a small percentage of random missingness (approximately 2%) is introduced to account for incidental data loss not captured by the MNAR corruption process.

Finally, a wrapper class for resolution augmentation was developed but was not used in the final experiments. This method was found to exacerbate existing missingness streaks, complicating model training. To handle varying sequence lengths, random cropping was applied to select sub-sequences for analysis.

## B  VQ-VAE ARCHITECTURAL DETAILS

The architectures for the three models (A0, A1, and A2) are illustrated in Figures 2b, 2c, and 2d, respectively. Throughout the network, spatial length was preserved to ensure that each time step—representing daily patient states—was captured in the embeddings.

For real-valued features such as *Sleep Start*, the mean squared error (MSE) loss was employed. This loss function was extended to continuous positive variables following the transformations described in Section 3. While the counting variables (*Location Clusters Count* and *Total Steps*) could be modeled using a Poisson distribution, the broad range of values ($15$ and $24,849$, respectively) allowed for an approximation using the MSE loss.

Binary features, such as *Weekend* and *Practiced Sport*, were trained using a modified binary cross-entropy (BCE) loss to account for class imbalances. Gradient norm clipping was applied, limiting the norm to a maximum of $2.0$ to ensure stable optimization and prevent gradient explosions in the early training phases, particularly for challenging variables such as *Location Distance*. The learning rate was initially set to $1 \times 10^{-3}$, with a learning rate scheduler (`ReduceLROnPlateau`) that applied a reduction factor of $0.1$ when no improvement was observed over 10 epochs.

The vector quantization (VQ) mechanism plays a key role in our architecture, particularly in models A1 and A2. A codebook of $256$ vectors, initialized randomly, was employed, with the embedding dimensionality set to $80$ for all variant architectures.

To combat the issue of codebook collapse—a common challenge in VQ-VAE models—a restart threshold of $0.1$ was applied. Embeddings that were underutilized (i.e., with utilization rates below this threshold) were re-initialized to improve code utilization following Dhariwal et al. (2020). This technique effectively mitigated collapse, as demonstrated by a monotonic increase in perplexity across training epochs. Both MCAR and MNAR experiments exhibited effective embedding utilization, which contributed to the overall performance.

As discussed in Section 3, our quantization mechanism leverages an exponential moving average (EMA) to update the embedding representations during training. This is controlled by a decay factor and the previously mentioned threshold that prevents underutilized embeddings from being excessively penalized. As part of the quantization step, a commitment loss is calculated to measure the difference between the input and its quantized representation, ensuring smooth transitions between different embeddings. For the experiments contained in this work, we used $\beta = 0.25$ in Equation 2.

To ensure the statistical rigor of our evaluation and to assess whether the observed differences between model variants are significant, we conducted a series of hypothesis tests. The analysis aims to determine whether the VQ-VAE model variants demonstrate statistically significant performance differences when compared to the baseline model A0, across various metrics. For more details, see Appendix E.1.

Model A0 serves as the baseline. It receives the zero-imputed signal as input, which is passed through four convolutional layers, each followed by batch normalization and a ReLU activation function. These layers use $3 \times 3$ filters with stride and padding set to 1, ensuring that the spatial dimensions are preserved. The encoder's output is then quantized using the VQ mechanism and passed to the decoder, which consists of four deconvolutional layers. Each deconvolutional layer is followed by batch normalization and ReLU, except for the last layer, where the identity function is applied to maintain the integrity of the output values for real-valued, continuous, and counting variables, and logits for binary variables. The complete architecture for model A0 can be seen in Table 3.

Model A1 incorporates the missingness mask alongside the zero-imputed signal. Prior to concatenation with the input signal, the mask undergoes processing through two convolutional layers, each followed by batch normalization and ReLU. After concatenation, the combined input is passed through six convolutional layers, similar to A0 but with additional depth to account for the mask information. The output is then quantized using the same VQ process, and the decoder operates identically to A0. The complete architecture for model A1 is described in Table 4.

Model A2 extends A1 by also passing the missingness mask to the decoder. The encoder processes the input identically to A1, quantizing the result before passing it to the decoder. In the decoder, the

quantized vector is processed alongside the mask, which is passed through two additional convolutional layers. These are followed by a block of four fine-tuning layers, which enable the decoder to integrate missingness information into the final reconstructed signal. The fine-tuning layers consist of convolutional layers followed by ReLU, except for the last layer, which uses the identity function. The complete architecture for model A2 is described in Table 5.

Table 3: Model A0 Architecture: Encoder, Quantizer, and Decoder

| Layer Type | Input Dimensions | Output Dimensions | Details |
|---|---|---|---|
| **Encoder** | | | |
| **Input (Signal)** | $[B, F, L]$ | - | Model input (signal) |
| Conv1D | $[B, F, L]$ | $[B, F, L]$ | $3 \times 3$, Stride = 1, Padding = 1 |
| BatchNorm1D | $[B, F, L]$ | $[B, F, L]$ | BatchNorm, after Conv1D |
| ReLU | $[B, F, L]$ | $[B, F, L]$ | Activation |
| Conv1D | $[B, F, L]$ | $[B, 2F, L]$ | $3 \times 3$, Stride = 1, Padding = 1 |
| BatchNorm1D | $[B, 2F, L]$ | $[B, 2F, L]$ | BatchNorm, after Conv1D |
| ReLU | $[B, 2F, L]$ | $[B, 2F, L]$ | Activation |
| Conv1D | $[B, 2F, L]$ | $[B, 4F, L]$ | $3 \times 3$, Stride = 1, Padding = 1 |
| BatchNorm1D | $[B, 4F, L]$ | $[B, 4F, L]$ | BatchNorm, after Conv1D |
| ReLU | $[B, 4F, L]$ | $[B, 4F, L]$ | Activation |
| Conv1D | $[B, 4F, L]$ | $[B, 8F, L]$ | $3 \times 3$, Stride = 1, Padding = 1 |
| BatchNorm1D | $[B, 8F, L]$ | $[B, 8F, L]$ | BatchNorm, after Conv1D |
| ReLU | $[B, 8F, L]$ | $[B, 8F, L]$ | Activation |
| **Quantizer** | | | |
| Quantization | $[B, 8F, L]$ | $[B, 8F, L]$ | VQ (Nearest Lookup) |
| **Decoder** | | | |
| Deconv1D | $[B, 8F, L]$ | $[B, 6F, L]$ | $3 \times 3$, Stride = 1, Padding = 1 |
| BatchNorm1D | $[B, 6F, L]$ | $[B, 6F, L]$ | BatchNorm, after Deconv1D |
| ReLU | $[B, 6F, L]$ | $[B, 6F, L]$ | Activation |
| Deconv1D | $[B, 6F, L]$ | $[B, 4F, L]$ | $3 \times 3$, Stride = 1, Padding = 1 |
| BatchNorm1D | $[B, 4F, L]$ | $[B, 4F, L]$ | BatchNorm, after Deconv1D |
| ReLU | $[B, 4F, L]$ | $[B, 4F, L]$ | Activation |
| Deconv1D | $[B, 4F, L]$ | $[B, 4F, L]$ | $3 \times 3$, Stride = 1, Padding = 1 |
| BatchNorm1D | $[B, 4F, L]$ | $[B, 4F, L]$ | BatchNorm, after Deconv1D |
| ReLU | $[B, 4F, L]$ | $[B, 4F, L]$ | Activation |
| Deconv1D | $[B, 4F, L]$ | $[B, 2F, L]$ | $3 \times 3$, Stride = 1, Padding = 1 |
| BatchNorm1D | $[B, 2F, L]$ | $[B, 2F, L]$ | BatchNorm, after Deconv1D |
| ReLU | $[B, 2F, L]$ | $[B, 2F, L]$ | Activation |
| Deconv1D | $[B, 2F, L]$ | $[B, F, L]$ | $3 \times 3$, Stride = 1, Padding = 1 |
| BatchNorm1D | $[B, F, L]$ | $[B, F, L]$ | BatchNorm, after Deconv1D |
| Identity | $[B, F, L]$ | $[B, F, L]$ | Model output: recons. value and logits (for binary) |

Table 4: Model A1 Architecture: Encoder, Quantizer, and Decoder

| Layer Type | Input Dimensions | Output Dimensions | Details |
|---|---|---|---|
| **Encoder** | | | |
| **Input (Signal)** | $[B, F, L]$ | - | Model input (signal) |
| **Input (Mask)** | $[B, M, L]$ | - | Model input (mask) |
| Conv1D (Mask) | $[B, M, L]$ | $[B, M, L]$ | $3 \times 3$, Stride = 1, Padding = 1 |
| BatchNorm1D (Mask) | $[B, M, L]$ | $[B, M, L]$ | BatchNorm, after Conv1D |
| ReLU (Mask) | $[B, M, L]$ | $[B, M, L]$ | Activation |
| Conv1D (Mask) | $[B, M, L]$ | $[B, M, L]$ | $3 \times 3$, Stride = 1, Padding = 1 |
| BatchNorm1D (Mask) | $[B, M, L]$ | $[B, M, L]$ | BatchNorm, after Conv1D |
| ReLU (Mask) | $[B, M, L]$ | $[B, M, L]$ | Activation |
| Concatenation (Signal + Mask) | $[B, F, L], [B, M, L]$ | $[B, F + M, L]$ | Concatenate signal and mask. Note: $F = M$ |
| Conv1D | $[B, F + M, L]$ | $[B, F, L]$ | $3 \times 3$, Stride = 1, Padding = 1 |
| BatchNorm1D | $[B, F, L]$ | $[B, F, L]$ | BatchNorm, after Conv1D |
| ReLU | $[B, F, L]$ | $[B, F, L]$ | Activation |
| Conv1D | $[B, F, L]$ | $[B, 2F, L]$ | $3 \times 3$, Stride = 1, Padding = 1 |
| BatchNorm1D | $[B, 2F, L]$ | $[B, 2F, L]$ | BatchNorm, after Conv1D |
| ReLU | $[B, 2F, L]$ | $[B, 2F, L]$ | Activation |
| Conv1D | $[B, 2F, L]$ | $[B, 4F, L]$ | $3 \times 3$, Stride = 1, Padding = 1 |
| BatchNorm1D | $[B, 4F, L]$ | $[B, 4F, L]$ | BatchNorm, after Conv1D |
| ReLU | $[B, 4F, L]$ | $[B, 4F, L]$ | Activation |
| Conv1D | $[B, 4F, L]$ | $[B, 4F, L]$ | $3 \times 3$, Stride = 1, Padding = 1 |
| BatchNorm1D | $[B, 4F, L]$ | $[B, 4F, L]$ | BatchNorm, after Conv1D |
| ReLU | $[B, 4F, L]$ | $[B, 4F, L]$ | Activation |
| Conv1D | $[B, 4F, L]$ | $[B, 6F, L]$ | $3 \times 3$, Stride = 1, Padding = 1 |
| BatchNorm1D | $[B, 6F, L]$ | $[B, 6F, L]$ | BatchNorm, after Conv1D |
| ReLU | $[B, 6F, L]$ | $[B, 6F, L]$ | Activation |
| Conv1D | $[B, 6F, L]$ | $[B, 8F, L]$ | $3 \times 3$, Stride = 1, Padding = 1 |
| BatchNorm1D | $[B, 8F, L]$ | $[B, 8F, L]$ | BatchNorm, after Conv1D |
| ReLU | $[B, 8F, L]$ | $[B, 8F, L]$ | Activation |
| **Quantizer** | | | |
| Quantization | $[B, 8F, L]$ | $[B, 8F, L]$ | VQ (Nearest Lookup) |
| **Decoder** | | | |
| Deconv1D | $[B, 8F, L]$ | $[B, 6F, L]$ | $3 \times 3$, Stride = 1, Padding = 1 |
| BatchNorm1D | $[B, 6F, L]$ | $[B, 6F, L]$ | BatchNorm, after Deconv1D |
| ReLU | $[B, 6F, L]$ | $[B, 6F, L]$ | Activation |
| Deconv1D | $[B, 6F, L]$ | $[B, 4F, L]$ | $3 \times 3$, Stride = 1, Padding = 1 |
| BatchNorm1D | $[B, 4F, L]$ | $[B, 4F, L]$ | BatchNorm, after Deconv1D |
| ReLU | $[B, 4F, L]$ | $[B, 4F, L]$ | Activation |
| Deconv1D | $[B, 4F, L]$ | $[B, 4F, L]$ | $3 \times 3$, Stride = 1, Padding = 1 |
| BatchNorm1D | $[B, 4F, L]$ | $[B, 4F, L]$ | BatchNorm, after Deconv1D |
| ReLU | $[B, 4F, L]$ | $[B, 4F, L]$ | Activation |
| Deconv1D | $[B, 4F, L]$ | $[B, 2F, L]$ | $3 \times 3$, Stride = 1, Padding = 1 |
| BatchNorm1D | $[B, 2F, L]$ | $[B, 2F, L]$ | BatchNorm, after Deconv1D |
| ReLU | $[B, 2F, L]$ | $[B, 2F, L]$ | Activation |
| Deconv1D | $[B, 2F, L]$ | $[B, F, L]$ | $3 \times 3$, Stride = 1, Padding = 1 |
| BatchNorm1D | $[B, F, L]$ | $[B, F, L]$ | BatchNorm, after Deconv1D |
| Identity | $[B, F, L]$ | $[B, F, L]$ | Model output: recons. value and logits (for binary) |

Table 5: Model A2 Architecture: Encoder, Quantizer, and Decoder

| Encoder | | | |
|---|---|---|---|
| **Layer Type** | **Input Dimensions** | **Output Dimensions** | **Details** |
| **Input (Signal)** | $[B, F, L]$ | - | Model input (signal) |
| **Input (Mask)** | $[B, M, L]$ | - | Model input (mask) |
| Conv1D (Mask) | $[B, M, L]$ | $[B, M, L]$ | $3 \times 3$, Stride = 1, Padding = 1 |
| BatchNorm1D (Mask) | $[B, M, L]$ | $[B, M, L]$ | BatchNorm, after Conv1D |
| ReLU (Mask) | $[B, M, L]$ | $[B, M, L]$ | Activation |
| Conv1D (Mask) | $[B, M, L]$ | $[B, M, L]$ | $3 \times 3$, Stride = 1, Padding = 1 |
| BatchNorm1D (Mask) | $[B, M, L]$ | $[B, M, L]$ | BatchNorm, after Conv1D |
| ReLU (Mask) | $[B, M, L]$ | $[B, M, L]$ | Activation |
| Concatenation (Signal + Mask) | $[B, F, L], [B, M, L]$ | $[B, F + M, L]$ | Concatenate signal and mask. Note: $F = M$ |
| Conv1D | $[B, F + M, L]$ | $[B, F, L]$ | $3 \times 3$, Stride = 1, Padding = 1 |
| BatchNorm1D | $[B, F, L]$ | $[B, F, L]$ | BatchNorm, after Conv1D |
| ReLU | $[B, F, L]$ | $[B, F, L]$ | Activation |
| Conv1D | $[B, F, L]$ | $[B, 2F, L]$ | $3 \times 3$, Stride = 1, Padding = 1 |
| BatchNorm1D | $[B, 2F, L]$ | $[B, 2F, L]$ | BatchNorm, after Conv1D |
| ReLU | $[B, 2F, L]$ | $[B, 2F, L]$ | Activation |
| Conv1D | $[B, 2F, L]$ | $[B, 4F, L]$ | $3 \times 3$, Stride = 1, Padding = 1 |
| BatchNorm1D | $[B, 4F, L]$ | $[B, 4F, L]$ | BatchNorm, after Conv1D |
| ReLU | $[B, 4F, L]$ | $[B, 4F, L]$ | Activation |
| Conv1D | $[B, 4F, L]$ | $[B, 4F, L]$ | $3 \times 3$, Stride = 1, Padding = 1 |
| BatchNorm1D | $[B, 4F, L]$ | $[B, 4F, L]$ | BatchNorm, after Conv1D |
| ReLU | $[B, 4F, L]$ | $[B, 4F, L]$ | Activation |
| Conv1D | $[B, 4F, L]$ | $[B, 6F, L]$ | $3 \times 3$, Stride = 1, Padding = 1 |
| BatchNorm1D | $[B, 6F, L]$ | $[B, 6F, L]$ | BatchNorm, after Conv1D |
| ReLU | $[B, 6F, L]$ | $[B, 6F, L]$ | Activation |
| Conv1D | $[B, 6F, L]$ | $[B, 8F, L]$ | $3 \times 3$, Stride = 1, Padding = 1 |
| BatchNorm1D | $[B, 8F, L]$ | $[B, 8F, L]$ | BatchNorm, after Conv1D |
| ReLU | $[B, 8F, L]$ | $[B, 8F, L]$ | Activation |
| Quantizer | | | |
| Quantization | $[B, 4F, L]$ | $[B, 4F, L]$ | VQ (Nearest Lookup) |
| Decoder | | | |
| **Input (Quantized Signal)** | $[B, 4F, L]$ | - | Model input (quantized signal) |
| **Input (Mask)** | $[B, M, L]$ | - | Model input (mask) |
| Conv1D (Mask) | $[B, M, L]$ | $[B, M, L]$ | $3 \times 3$, Stride = 1, Padding = 1 |
| BatchNorm1D (Mask) | $[B, M, L]$ | $[B, M, L]$ | BatchNorm, after Conv1D |
| ReLU (Mask) | $[B, M, L]$ | $[B, M, L]$ | Activation |
| Conv1D (Mask) | $[B, M, L]$ | $[B, M, L]$ | $3 \times 3$, Stride = 1, Padding = 1 |
| BatchNorm1D (Mask) | $[B, M, L]$ | $[B, M, L]$ | BatchNorm, after Conv1D |
| ReLU (Mask) | $[B, M, L]$ | $[B, M, L]$ | Activation |
| Deconv1D (Signal) | $[B, 8F, L]$ | $[B, 6F, L]$ | $3 \times 3$, Stride = 1, Padding = 1 |
| BatchNorm1D (Signal) | $[B, 6F, L]$ | $[B, 6F, L]$ | BatchNorm, after Deconv1D |
| ReLU (Signal) | $[B, 6F, L]$ | $[B, 6F, L]$ | Activation |
| Deconv1D (Signal) | $[B, 6F, L]$ | $[B, 4F, L]$ | $3 \times 3$, Stride = 1, Padding = 1 |
| BatchNorm1D (Signal) | $[B, 4F, L]$ | $[B, 4F, L]$ | BatchNorm, after Deconv1D |
| ReLU (Signal) | $[B, 4F, L]$ | $[B, 4F, L]$ | Activation |
| Deconv1D (Signal) | $[B, 4F, L]$ | $[B, 4F, L]$ | $3 \times 3$, Stride = 1, Padding = 1 |
| BatchNorm1D (Signal) | $[B, 4F, L]$ | $[B, 4F, L]$ | BatchNorm, after Deconv1D |
| ReLU (Signal) | $[B, 4F, L]$ | $[B, 4F, L]$ | Activation |
| Deconv1D (Signal) | $[B, 4F, L]$ | $[B, 2F, L]$ | $3 \times 3$, Stride = 1, Padding = 1 |
| BatchNorm1D (Signal) | $[B, 2F, L]$ | $[B, 2F, L]$ | BatchNorm, after Deconv1D |
| ReLU (Signal) | $[B, 2F, L]$ | $[B, 2F, L]$ | Activation |
| Deconv1D (Signal) | $[B, 2F, L]$ | $[B, F, L]$ | $3 \times 3$, Stride = 1, Padding = 1 |
| BatchNorm1D (Signal) | $[B, F, L]$ | $[B, F, L]$ | BatchNorm, after Deconv1D |
| ReLU (Signal) | $[B, F, L]$ | $[B, F, L]$ | Activation |
| Concatenation (Quantized Signal + Mask) | $[B, F, L], [B, M, L]$ | $[B, F + M, L]$ | Concatenate signal and mask. Note: $F = M$ |
| Fine-tuning Conv1D | $[B, F + M, L]$ | $[B, F + M, L]$ | $3 \times 3$, Stride = 1, Padding = 1 |
| BatchNorm1D | $[B, F + M, L]$ | $[B, F + M, L]$ | BatchNorm, after Conv1D |
| ReLU | $[B, F + M, L]$ | $[B, F + M, L]$ | Activation |
| Fine-tuning Conv1D | $[B, F + M, L]$ | $[B, F, L]$ | $3 \times 3$, Stride = 1, Padding = 1 |
| BatchNorm1D | $[B, F, L]$ | $[B, F, L]$ | BatchNorm, after Conv1D |

| Encoder (continued) | | | |
|---|---|---|---|
| **Layer Type** | **Input Dimensions** | **Output Dimensions** | **Details** |
| ReLU | $[B, F, L]$ | $[B, F, L]$ | Activation |
| Fine-tuning Conv1D | $[B, F, L]$ | $[B, F, L]$ | $3 \times 3$, Stride = 1, Padding = 1 |
| BatchNorm1D | $[B, F, L]$ | $[B, F, L]$ | BatchNorm, after Conv1D |
| ReLU | $[B, F, L]$ | $[B, F, L]$ | Activation |
| Fine-tuning Conv1D | $[B, F, L]$ | $[B, F, L]$ | $3 \times 3$, Stride = 1, Padding = 1 |
| BatchNorm1D | $[B, F, L]$ | $[B, F, L]$ | BatchNorm, after Conv1D |
| Identity | $[B, F, L]$ | $[B, F, L]$ | Model output: recons. value and logits (for binary) |

## C    CONSTRUCTING VQ-VAE LATENT PROFILES FOR CPD

In preparing VQ-VAE profiles for use in the CPD task, we leverage the inherent sparsity of the learned representations. This sparsity not only enhances the interpretability of the patient time-series embeddings but also allows for efficient and accurate change-point detection, critical in real-world applications for patient behavior monitoring for psychiatric patients.

VQ-VAE representations often exhibit significant variations in the frequency of usage across embeddings. To capitalize on this, we introduce a ranking system based on the frequency of each embedding's occurrence. Embeddings that appear frequently within the time-series sample are ranked higher, as these are likely to represent more common patterns. Conversely, embeddings that are infrequently used (below a certain number of "most used embeddings") are considered outliers and grouped into a special category referred to as the "dummy" embedding. This dummy embedding is more than a placeholder; it reflects rare or anomalous patterns, which may acquire specific clinical interpretations, such as periods of abnormal patient behavior or sensor malfunction. In particular, for the CPD results shown in Figure 5, only a small number of individual embeddings ranging from 5 to 30 (depending on the specific setting)—out of the total 256 in the codebook—were considered, with the remaining, less-used instances being classified into the "dummy" embedding. A detailed discussion on the number of individual profiles used can be found in Section 4.2 and ablation study regarding the number of individual embeddings considered for the CPD algorithm is provided in Appendix D.

By categorizing uncommon embeddings into a collective representation, we enhance the robustness of downstream analysis, as this method mitigates the noise introduced by outlier embeddings (themselves caused by outlier, and often erroneous, data) while retaining the capacity to detect important deviations in patient behavior.

As mentioned in Section 4, CPD can be approached in both deterministic and probabilistic modes, depending on the level of certainty required in detecting shifts in patient behavior. To support both approaches, we compute pseudo-probabilities derived from the distances between the quantized embeddings and the original continuous outputs of the encoder. Since the latent space of VQ-VAE is discrete, pseudo-probabilities are computed by first calculating the Euclidean distances between the continuous encoder outputs and the set of embeddings in the latent space. These distances quantify how close or far each input is from each embedding. Next, the softmax function is applied to the additive inverse of these distances, transforming them into a probability distribution over all possible embeddings. This transformation ensures that embeddings closer to the continuous encoder output (i.e., those with smaller Euclidean distances) are assigned higher pseudo-probabilities, while more distant embeddings are assigned lower pseudo-probabilities, thereby approximating a probabilistic interpretation for the otherwise discrete latent profiles.

These probabilities provide a soft assignment, offering an interpretable measure of how well an embedding fits the original data point. This is particularly useful in probabilistic CPD, where transitions between states are inherently uncertain, and the distances can be used to modulate the likelihood of a change-point. By integrating both deterministic hard-assignments and probabilistic soft-assignments, our framework allows for flexible CPD that can adapt to different levels of interpretability and precision, essential for clinical scenarios.

## D CPD ALGORITHM DETAILS AND ABLATION STUDY

The change-point detector (CPD) model used in this work was designed with many customization options, including CPD versions, hyperparameters, and alternative methods. Some of these options are explained in detail next.

The most important setting in the CPD is whether to use the hierarchical version (Moreno-Muñoz et al., 2019), which is designed to accept profile sequences of discrete nature, or the multinomial CPD presented in (Romero-Medrano et al., 2022) that has been adapted to work with profile distributions, which provide a richer characterization of the latent representation.

- **Hierarchical CPD**. As explained in Section 4.2, instead of directly analyzing the high-dimensional observations, the hierarchical CPD is fed with a latent variable (one discrete profile per day) and infers the posterior distribution of changes in such pseudo-observations. This approach simplifies the detection process and reduces computational complexity. However, when the distributions of the latent variables are flat or uncertain, the hierarchical CPD's performance can be compromised due to noisy point estimates (i.e., the categorical estimation of the profiles is not modeled with confidence).

- **Multinomial CPD**. The multinomial CPD addresses this limitation by incorporating multinomial sampling to better characterize the uncertainty in latent variable inference. Instead of relying solely on point estimates, the multinomial CPD draws multiple samples from the posterior distribution of latent variables at each time step and constructs a counting vector representing the frequency of each latent class within the samples. By considering the uncertainty in latent variable inference, the multinomial CPD improves detection rate and enhances robustness to noisy or missing data.

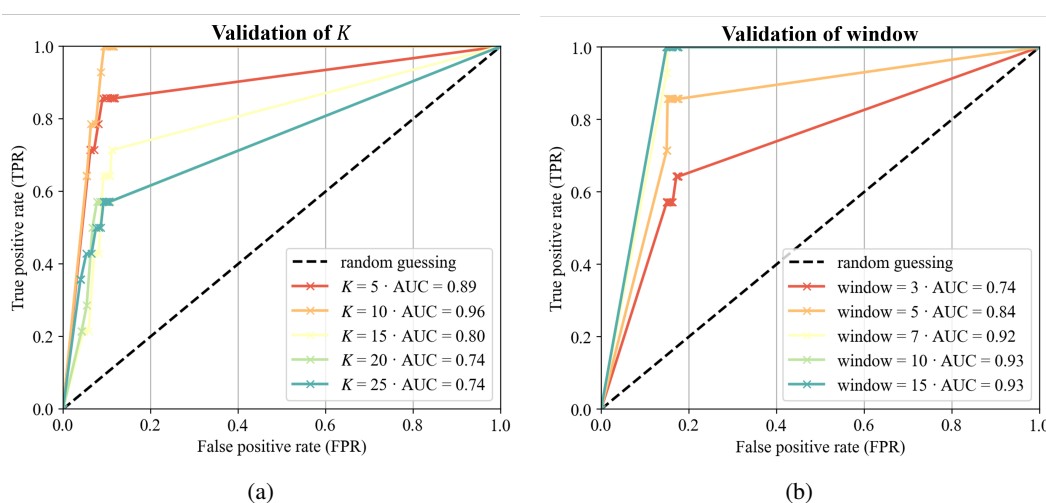

(a)  (b)

Figure 6: ROC curves obtained from a hyperparameter analysis on the HetMM–CPD integration, testing a range of values of (a) the number of profiles $K$ and (b) the size of the temporal window. The configuration of the baseline HetMM–CPD pipeline used as reference was set to 10 profiles (the best-performing value) and a 7-day window size.

Some of the hyperparameters involved in the downstream task were fixed based on our previous experience working with the HetMM–CPD pipeline. A brief description is given for each of them:

- **Number of profiles,** $K$. While not a hyperparameter of the CPD stage (but rather involved in the VQ-VAE or HetMM steps), the number of possible profiles is a crucial setting in the downstream task. Too few profiles will fail to capture the distinct behavior patterns, but too many may introduce noisy profiles modeled with low confidence that impede the correct performance of the CPD. The value of $K$ in the heterogeneous mixture model was analyzed (Figure 6a) and chosen to be 10.

- **Number of samples in multinomial distribution,** $S$. In the multinomial approach, $S$ represents the number of samples that are drawn from the posterior distribution of the latent variables at each time step. A larger value will adapt better to the latent profiles but also complicates the detection task of the CPD. The results provided in Section 5.2 were obtained with $S = 10$.

- **Prior change-point probability,** $\lambda$. As explained in Section 4.1, $\lambda$ is involved in the hazard function that defines the prior probability of having a change-point at any instant. This constant can be tuned to adapt the CPD's sensitivity and a few values were included in the results offered in Figure 5 of Section 5.2.

- **Size of the temporal window,** $w$. The CPD model focuses on a temporal frame to assess whether its predictions are successful or not. For example, for each true event, a true positive is returned if an alarm was given by the model within the temporal window previous to that event. If the CPD did not predict any change, then a false negative is counted. This window hyperparameter allows therefore to select how long in advance we aim to predict suicide events. We chose a prediction period of one week ($w = 7$ days), which obtained a high AUC in our analysis (see Figure 6b) and is brief enough to serve as short-term prediction.

- **Threshold,** $\tau$. The last hyperparameter affects the definition of alarms or positive predictions (i.e., the conversion from run length to a binary detection vector). Three methods are implemented in the CPD model. The first one, named *MAP ratio*, was used in this work.

  - *MAP ratio* (default) $\rightarrow$ based on the MAP estimates of the run length, an alarm is returned if the ratio of current $r_t$ over the previous day $r_{t-1}$ is below the threshold:
  
  $$\frac{r_t}{r_{t-1}} < \tau$$
  
  - *MAP difference* $\rightarrow$ based on the MAP estimates of the run length, an alarm is returned if the difference between current $r_t$ and previous $r_{t-1}$ is above the threshold:
  
  $$r_t - r_{t-1} > \tau$$
  
  - *Cumulative sum* $\rightarrow$ based on the cumulative probability of the run length of previous days (within the specified window of size $w$), an alarm is returned if this sum is above the threshold:
  
  $$\sum_{i=0}^{w} r_{t-i} > \tau$$

Regarding the incorporation of the VQ-VAE encoded space as input to the CPD, we tested the different model types A0, A1 and A2 explained in Appendix B, and for a range of numbers of embeddings (i.e., the number of possible profiles used in the subject characterization, $K$). The results are displayed in Figure 7. These graphs were obtained using the VQ-VAE's discrete profiles, not their pseudo-probabilities. The three VQ-VAE model variations yielded similar results, with version A1 often reaching a 100% of sensitivity. In the case of models A0 and A2, performance depended heavily on the value of $K$, with poorer outcomes when less profiles were used ($K = 5$, $K = 10$). The optimum number of profiles seemed to be 20, a reason why this value would be used to produce Figures 5b and 5c in the results section.

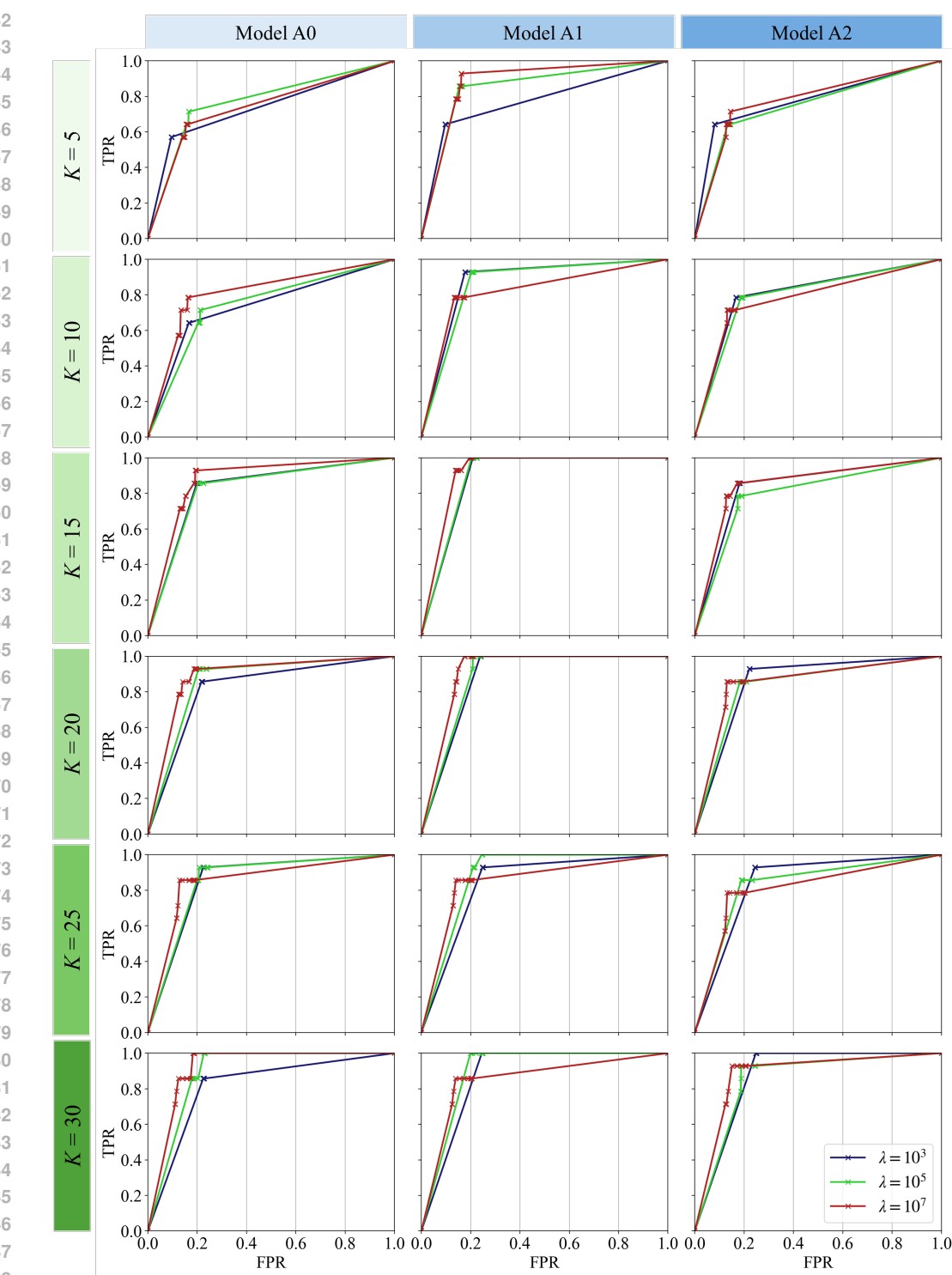

Figure 7: ROC curves resulting of the VQ-VAE–CPD integration using discrete profiles. The figure compares models A0, A1 and A2 (columns) and different numbers of embeddings or profiles $K$ (rows). The three colored lines in each plot correspond to three different values of hyperparameter $\lambda$.

# E    EXTENDED RESULTS ON THE VQ-VAE FOUNDATION MODEL

## E.1    SIGNAL RECONSTRUCTION AND IMPUTATION

Table 6 presents the reconstruction performance in terms of MAE (or F1 score for the binary variables *Weekend* and *Practice Sport*) for observed data, as well as for missing data under both MCAR and MNAR mechanisms. The results indicate that all three models perform comparably across most variables, with some nuanced differences. For example, Model A2 performs better on reconstructing observed instances of *Sleep Start*, achieving lower Mean Absolute Error (MAE) compared to A0 and A1. Conversely, Models A0 and A1 perform better than A2 for reconstructing observed instances of *Time at Home* and *Sleep Duration*. Additionally, A0 achieves the lowest error for the observed instances of *Total Steps*.

Despite not being explicitly optimized for imputation, the models performed competently in this task. These results highlight the models' ability to generalize beyond their training objective, particularly under the MNAR condition, where missingness is more structured and challenging. This is compounded by the fact that the discrete profile representation provided by VQ-VAE is sparse, i.e., out of the total 256 embeddings in the codebook, only a few were used for each patient, thereby enhancing interpretability (see Appendix E.2 for embedding utilization histograms).

It is important to note that no synthetic missingness was applied to the variables *Weekend* and *Practiced Sport*, as these were fully observed across the dataset. Consequently, the MCAR and MNAR scenarios were not applicable for these variables. Nonetheless, the consistently high F1 scores (close to 1.0) achieved by all models for these categorical variables reinforce the robustness of the learned representations, even for variables without missing data.

Hypothesis testing was performed for a more in-depth analysis to assess the statistical significance of the observed differences between the models. We began by testing the normality of the data using the Shapiro-Wilk test. The null hypothesis ($H_0$) for this test states that the data comes from a normally distributed population. Conversely, the alternative hypothesis ($H_1$) posits that the data is not normally distributed. We employed a significance level of $\alpha = 0.05$. If the $p$-value from the Shapiro-Wilk test is greater than $0.05$, we fail to reject the null hypothesis, indicated that the data can be assumed to follow a normal distribution.[7]

The Shapiro-Wilk test results are provided in Table 7. If both models' result (i.e., the variant model and baseline A0) for a given variable and type passed the normality test, we proceeded with the paired Welch t-test. If the null hypothesis was rejected for either one of the two models (i.e., the data is not normally distributed), we opted for the non-parametric Wilcoxon signed-rank test.

When the data for both the baseline and the variant model were found to be normally distributed, we used the paired Welch's t-test to compare their means. The null hypothesis for this test asserts that there is not difference between the means of the two models, while the alternative hypothesis suggests a significant difference between them. We again used a significance level of $\alpha = 0.05$, rejecting the null hypothesis if the p-value was below this threshold. The results for the paired Welch t-tests are summarized in Table 8.

For cases where the data for one or both models did not pass the Shapiro-Wilk normality test, we employed the Wilcoxon signed-rank test. This non-parametric test does not assume normality.[8] The null hypothesis here is that the distributions of the two models are identical, while the alternative hypothesis suggests a significant difference between them. Similar to the Welch t-test, we used $\alpha = 0.05$ as the significance level. Table 9 provides a detailed summary of the Wilcoxon signed-rank test results.

Figure 8 and Figure 9 present reconstructed and imputed sample examples, where white shading indicates observed data, grey shading denotes originally missing data, and purple shading represents synthetically induced missingness. The remaining time steps (in this case, days) are fully visible to the model. When the original signal is obscured in observed intervals, it is due to one or more model

---

[7]The significance levels used in these tests ensure that any rejection of the null hypothesis corresponds to a less than 5% probability of a Type I error, i.e., that it is rejected while being true. In the case of the Shapiro-Wilk and Wilcoxon signed-rank tests this would represent the scenario in which it is incorrectly concluded that the models differ when they do not.

[8]A requirement of the Wilcoxon signed-rank test is symmetry.

reconstructions perfectly overlapping the true signal, demonstrating accurate recovery. As shown in Figure 8a and Figure 9a all models perform well with binary variables. Notably, the proposed VQ-VAE variants exhibit strong imputation capabilities even under high proportions of missingness, as evidenced by Figure 8c, Figure 8f, and Figure 9e. Whether the missing data spans large temporal segments (e.g., the first three-quarters of the sample in Figure 8f), appears centrally (Figure 9g), or is intermittently distributed (Figure 8d), the models consistently maintain robust representations and plausible imputations. This performance generalizes across all variable types—continuous real-valued, continuous positive, count data, and binary—highlighting the versatility of the models across different data ranges and types.

## E.2 EMBEDDING USAGE HISTOGRAMS

The discrete quantization of VQ-VAE facilitates the construction of latent representations, making it particularly suited for applications that benefit from codifying instances, as demonstrated in this work. Unlike traditional methods that rely on handcrafted features—often tailored to individual patients and limiting generalizability—VQ-VAE learns patient-agnostic embeddings, enabling generalization across subpopulations and tasks. These discrete embeddings can be effectively applied to tasks such as time-series data imputation and extended to critical downstream tasks, such as identifying critical health events or suicide risk detection. As illustrated in Figure 10, the usefulness of these embeddings is enhanced by their sparsity—typically, only a small subset of the 256 available embeddings is used per sample. This results in a more interpretable solution, with infrequent embeddings classified as "dummy" embeddings, which can themselves acquire meaningful interpretations (e.g., representing rare or unstable states). In turn, this sparsity in then leveraged to provide contained, yet expressive profiles sequences for the CPD algorithm, as discussed in Appendix C.

Table 6: Performance of Models A0, A1, and A2. Metrics for Variables 0-7 are reported in MAE (lower is better), and Variables 8-9 are evaluated using F1 (higher is better).

| Variable | Type | Model A0 | Model A1 | Model A2 |
|---|---|---|---|---|
| **Sleep Start (s)** | XO | $1315.63 \pm 47.06$ | $1242.66 \pm 57.88$ | $\mathbf{1177.78 \pm 57.75}$ |
| | MCAR | $5777.24 \pm 229.41$ | $5651.99 \pm 245.31$ | $5578.96 \pm 496.26$ |
| | MNAR | $5896.85 \pm 492.96$ | $5718.97 \pm 417.62$ | $5607.64 \pm 593.95$ |
| **Traveled Distance (m)** | XO | $12202.43 \pm 1296.66$ | $11627.66 \pm 937.86$ | $12874.13 \pm 836.27$ |
| | MCAR | $17008.33 \pm 7488.46$ | $16681.98 \pm 13920.55$ | $15190.03 \pm 3520.84$ |
| | MNAR | $15100.38 \pm 2035.91$ | $14232.06 \pm 1821.58$ | $15175.21 \pm 2363.39$ |
| **Time at Home (m)** | XO | $\mathbf{146.17 \pm 4.95}$ | $\mathbf{143.58 \pm 8.58}$ | $174.94 \pm 9.70$ |
| | MCAR | $289.52 \pm 17.03$ | $290.18 \pm 17.87$ | $291.85 \pm 18.18$ |
| | MNAR | $287.52 \pm 16.05$ | $282.68 \pm 15.94$ | $286.16 \pm 13.35$ |
| **Sleep Duration (s)** | XO | $\mathbf{4149.40 \pm 120.98}$ | $\mathbf{4055.13 \pm 151.20}$ | $5005.76 \pm 211.03$ |
| | MCAR | $6563.44 \pm 282.73$ | $6615.74 \pm 309.10$ | $6738.00 \pm 398.30$ |
| | MNAR | $6422.58 \pm 340.45$ | $6373.11 \pm 232.31$ | $6585.21 \pm 300.78$ |
| **Time Walking (s)** | XO | $1341.44 \pm 65.39$ | $1298.03 \pm 61.20$ | $1279.72 \pm 67.14$ |
| | MCAR | $1779.98 \pm 145.89$ | $1742.47 \pm 101.91$ | $1734.54 \pm 73.66$ |
| | MNAR | $1676.90 \pm 82.56$ | $1657.30 \pm 96.37$ | $1744.46 \pm 105.72$ |
| **App Usage Total (s)** | XO | $3784.17 \pm 348.70$ | $3714.48 \pm 315.91$ | $3968.00 \pm 357.25$ |
| | MCAR | $5045.95 \pm 528.72$ | $4973.86 \pm 558.61$ | $4946.72 \pm 744.72$ |
| | MNAR | $4436.77 \pm 669.15$ | $4303.00 \pm 760.17$ | $4310.54 \pm 655.41$ |
| **Location Clusters Count** | XO | $1.0887 \pm 0.0716$ | $1.0746 \pm 0.0833$ | $1.2469 \pm 0.0987$ |
| | MCAR | $1.3234 \pm 0.1120$ | $1.3143 \pm 0.1094$ | $1.3980 \pm 0.1100$ |
| | MNAR | $1.3210 \pm 0.1887$ | $1.2900 \pm 0.1907$ | $1.3835 \pm 0.1645$ |
| **Total Steps** | XO | $\mathbf{2101.48 \pm 348.70}$ | $3714.48 \pm 315.91$ | $3968.00 \pm 357.25$ |
| | MCAR | $3056.67 \pm 137.87$ | $3002.53 \pm 230.60$ | $2993.74 \pm 204.87$ |
| | MNAR | $3042.64 \pm 130.44$ | $2986.37 \pm 175.30$ | $2986.15 \pm 164.41$ |
| **Weekend** | XO | $0.9950 \pm 0.0010$ | $0.9960 \pm 0.0015$ | $0.9967 \pm 0.0013$ |
| **Practiced Sport** | XO | $0.9932 \pm 0.0016$ | $0.9941 \pm 0.0023$ | $0.9929 \pm 0.0021$ |

Table 7: Shapiro-Wilk test for normality for models A0, A1, and A2. The table reports the test statistic (W) and p-values for each model and variable under different conditions (XO, MCAR, and MNAR). $\alpha = 0.05$ was used and ✗ denotes the rejection of the null at the $\alpha$ significance level, implying non-normal distribution.

| Variable | Condition | Model A0 (W) | Model A0 (p) | Model A1 (W) | Model A1 (p) | Model A2 (W) | Model A2 (p) |
|---|---|---|---|---|---|---|---|
| **Sleep Start (s)** | XO | 0.9870 | 0.9197 | 0.9515 | 0.0854 | 0.9639 | 0.2274 |
| | MCAR | 0.9654 | 0.2542 | 0.9877 | 0.9358 | 0.9758 | 0.5371 |
| | MNAR | 0.9544 | 0.1074 | 0.9352 | 0.0240 (✗) | 0.9839 | 0.8290 |
| **Traveled Distance (m)** | XO | 0.7935 | $5 \times 10^{-6}$ (✗) | 0.9768 | 0.5723 | 0.9827 | 0.7863 |
| | MCAR | 0.4596 | $5.9 \times 10^{-11}$ (✗) | 0.2506 | $5 \times 10^{-13}$ (✗) | 0.4973 | $1.6 \times 10^{-10}$ (✗) |
| | MNAR | 0.9714 | 0.3969 | 0.9756 | 0.5311 | 0.9748 | 0.5023 |
| **Time at Home (m)** | XO | 0.9645 | 0.2387 | 0.9537 | 0.1016 | 0.9589 | 0.1530 |
| | MCAR | 0.9862 | 0.8978 | 0.9402 | 0.0351 (✗) | 0.9700 | 0.3595 |
| | MNAR | 0.9668 | 0.2833 | 0.9604 | 0.1734 | 0.9576 | 0.1387 |
| **Sleep Duration (s)** | XO | 0.9720 | 0.4141 | 0.9548 | 0.1113 | 0.9639 | 0.2270 |
| | MCAR | 0.9658 | 0.2636 | 0.9640 | 0.2292 | 0.9803 | 0.7008 |
| | MNAR | 0.9654 | 0.2545 | 0.9782 | 0.6245 | 0.9484 | 0.0668 |
| **Time Walking (s)** | XO | 0.9682 | 0.3155 | 0.9617 | 0.1913 | 0.9706 | 0.3751 |
| | MCAR | 0.7455 | $5.9 \times 10^{-7}$ (✗) | 0.9734 | 0.4593 | 0.9868 | 0.9138 |
| | MNAR | 0.9747 | 0.4988 | 0.8987 | 0.0017 (✗) | 0.9864 | 0.9046 |
| **App Usage Total (s)** | XO | 0.9629 | 0.2106 | 0.9611 | 0.1821 | 0.9596 | 0.1620 |
| | MCAR | 0.9700 | 0.3602 | 0.9782 | 0.6242 | 0.7979 | $6.1 \times 10^{-6}$ (✗) |
| | MNAR | 0.9259 | 0.0119 (✗) | 0.9248 | 0.010 (✗) | 0.9733 | 0.4549 |
| **Location Clusters Count** | XO | 0.9576 | 0.1386 | 0.9642 | 0.2321 | 0.9838 | 0.8272 |
| | MCAR | 0.9754 | 0.5245 | 0.9567 | 0.1290 | 0.9443 | 0.0487 (✗) |
| | MNAR | 0.9612 | 0.1841 | 0.9717 | 0.4063 | 0.9742 | 0.4836 |
| **Total Steps** | XO | 0.9574 | 0.1366 | 0.9696 | 0.3496 | 0.9790 | 0.6536 |
| | MCAR | 0.9745 | 0.4929 | 0.9057 | 0.0028 (✗) | 0.9232 | 0.0097 (✗) |
| | MNAR | 0.9800 | 0.6911 | 0.9818 | 0.7552 | 0.9487 | 0.0683 |
| **Weekend** | XO | 0.9849 | 0.9849 | 0.9752 | 0.5162 | 0.9617 | 0.9617 |
| **Practiced Sport** | XO | 0.9397 | 0.0338 (✗) | 0.7819 | $2.9 \times 10^{-6}$ (✗) | 0.9503 | 0.0779 |

Table 8: Paired Welch's t-test results comparing model variant models A1 and A2 to the baseline (A0). The table reports the test statistic (t) and p-values for each model and variable under different conditions (XO, MCAR, and MNAR). $\alpha = 0.05$ was used and ✗ denotes the rejection of the null hypothesis at the $\alpha$ significance level.

| Variable | Condition | A0 vs A1 ($t$) | A0 vs A1 ($p$) | A0 vs A2 ($t$) | A0 vs A2 ($p$) |
|---|---|---|---|---|---|
| **Sleep Start (s)** | XO | $-6.1860$ | $3 \times 10^{-8}$ (✗) | $-11.7016$ | $1.4 \times 10^{-18}$ (✗) |
| | MCAR | $-2.3585$ | 0.0209 (✗) | $-2.2937$ | 0.0257 (✗) |
| | MNAR | — | — | $-2.3697$ | 0.0203 (✗) |
| **Traveled Distance (m)** | XO | — | — | — | — |
| | MCAR | — | — | — | — |
| | MNAR | $-2.0102$ | 0.0479 (✗) | 0.1517 | 0.8798 |
| **Time at Home (m)** | XO | $-1.6511$ | 0.1037 | 16.7191 | $7.4 \times 10^{-24}$ (✗) |
| | MCAR | — | — | 0.5906 | 0.5564 |
| | MNAR | $-1.0755$ | 0.2854 | $-0.4124$ | 0.6812 |
| **Sleep Duration (s)** | XO | $-3.0788$ | 0.0029 (✗) | 22.2654 | $2.6 \times 10^{-31}$ (✗) |
| | MCAR | 0.7896 | 0.4322 | 2.2603 | 0.0268 (✗) |
| | MNAR | $-0.7592$ | 0.4503 | 2.2641 | 0.0264 (✗) |
| **Time Walking (s)** | XO | $-3.0425$ | 0.0031 (✗) | $-4.1449$ | $8.6 \times 10^{-5}$ (✗) |
| | MCAR | — | — | — | — |
| | MNAR | — | — | 3.1853 | 0.0021 (✗) |
| **App Usage Total (s)** | XO | $-0.9368$ | 0.3518 | 2.3289 | 0.0225 (✗) |
| | MCAR | $-0.5927$ | 0.5551 | — | — |
| | MNAR | — | — | — | — |
| **Location Clusters Count** | XO | $-0.8132$ | 0.4186 | 8.2048 | $6.9 \times 10^{-12}$ (✗) |
| | MCAR | $-0.3650$ | 0.7160 | — | — |
| | MNAR | $-0.7398$ | 0.4616 | 1.5771 | 0.1189 |
| **Total Steps** | XO | $-0.1357$ | 0.8924 | 5.2860 | $1.1 \times 10^{-6}$ (✗) |
| | MCAR | — | — | — | — |
| | MNAR | $-1.6286$ | 0.1078 | $-1.7023$ | 0.0929 |
| **Weekend** | XO | 3.6438 | 0.0005 (✗) | 6.3882 | $1.5 \times 10^{-8}$ (✗) |
| **Practiced Sport** | XO | — | — | — | — |

Table 9: Wilcoxon signed-rank test results comparing model variant models A1 and A2 to the baseline (A0). The table reports the test statistic (t) and p-values for each model and variable under different conditions (XO, MCAR, and MNAR). $\alpha = 0.05$ was used and ✗ denotes the rejection of the null hypothesis at the $\alpha$ significance level.

| Variable | Condition | A0 vs A1 ($t$) | A0 vs A1 ($p$) | A0 vs A2 ($t$) | A0 vs A2 ($p$) |
|---|---|---|---|---|---|
| **Sleep Start (s)** | XO | — | — | — | — |
| | MCAR | — | — | — | — |
| | MNAR | 272.0 | 0.0641 | — | — |
| **Traveled Distance (m)** | XO | 217.0 | 0.0086 (✗) | 200.0 | 0.0041 (✗) |
| | MCAR | 263.0 | 0.0482 (✗) | 353.0 | 0.4517 |
| | MNAR | — | — | — | — |
| **Time at Home (m)** | XO | — | — | — | — |
| | MCAR | 394.0 | 0.8368 | — | — |
| | MNAR | — | — | — | — |
| **Sleep Duration (s)** | XO | — | — | — | — |
| | MCAR | — | — | — | — |
| | MNAR | — | — | — | — |
| **Time Walking (s)** | XO | — | — | — | — |
| | MCAR | 333.0 | 0.3074 | 310.0 | 0.1831 |
| | MNAR | 301.0 | 0.1461 | — | — |
| **App Usage Total (s)** | XO | — | — | — | — |
| | MCAR | — | — | 301.0 | 0.1460 |
| | MNAR | 330.0 | 0.2887 | 369.0 | 0.5900 |
| **Location Clusters Count** | XO | — | — | — | — |
| | MCAR | — | — | 206.0 | 0.0053 |
| | MNAR | — | — | — | — |
| **Total Steps** | XO | — | — | — | — |
| | MCAR | 283.0 | 0.0892 | 280.0 | 0.0817 |
| | MNAR | — | — | — | — |
| **Weekend** | XO | — | — | — | — |
| **Practiced Sport** | XO | 236.0 | 0.0185 (✗) | 353.0 | 0.5360 |

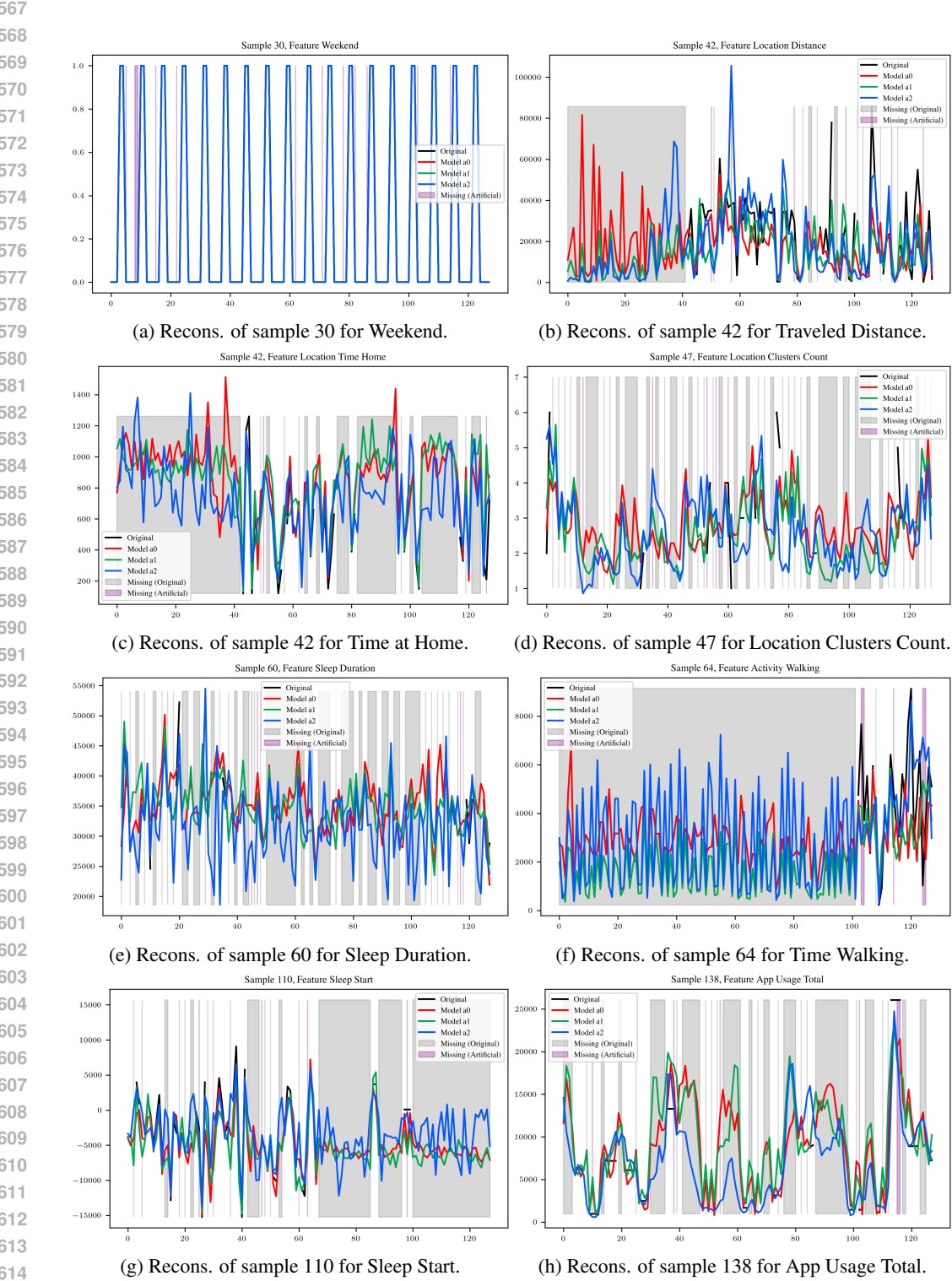

(a) Recons. of sample 30 for Weekend.

(b) Recons. of sample 42 for Traveled Distance.

(c) Recons. of sample 42 for Time at Home.

(d) Recons. of sample 47 for Location Clusters Count.

(e) Recons. of sample 60 for Sleep Duration.

(f) Recons. of sample 64 for Time Walking.

(g) Recons. of sample 110 for Sleep Start.

(h) Recons. of sample 138 for App Usage Total.

Figure 8: Representative signal reconstructions for observed and imputed instances. In cases where the original signal is not explicitly shown, it is because one or more of the models (whose reconstructions are plotted) overlap the true signal precisely, obscuring the original data.

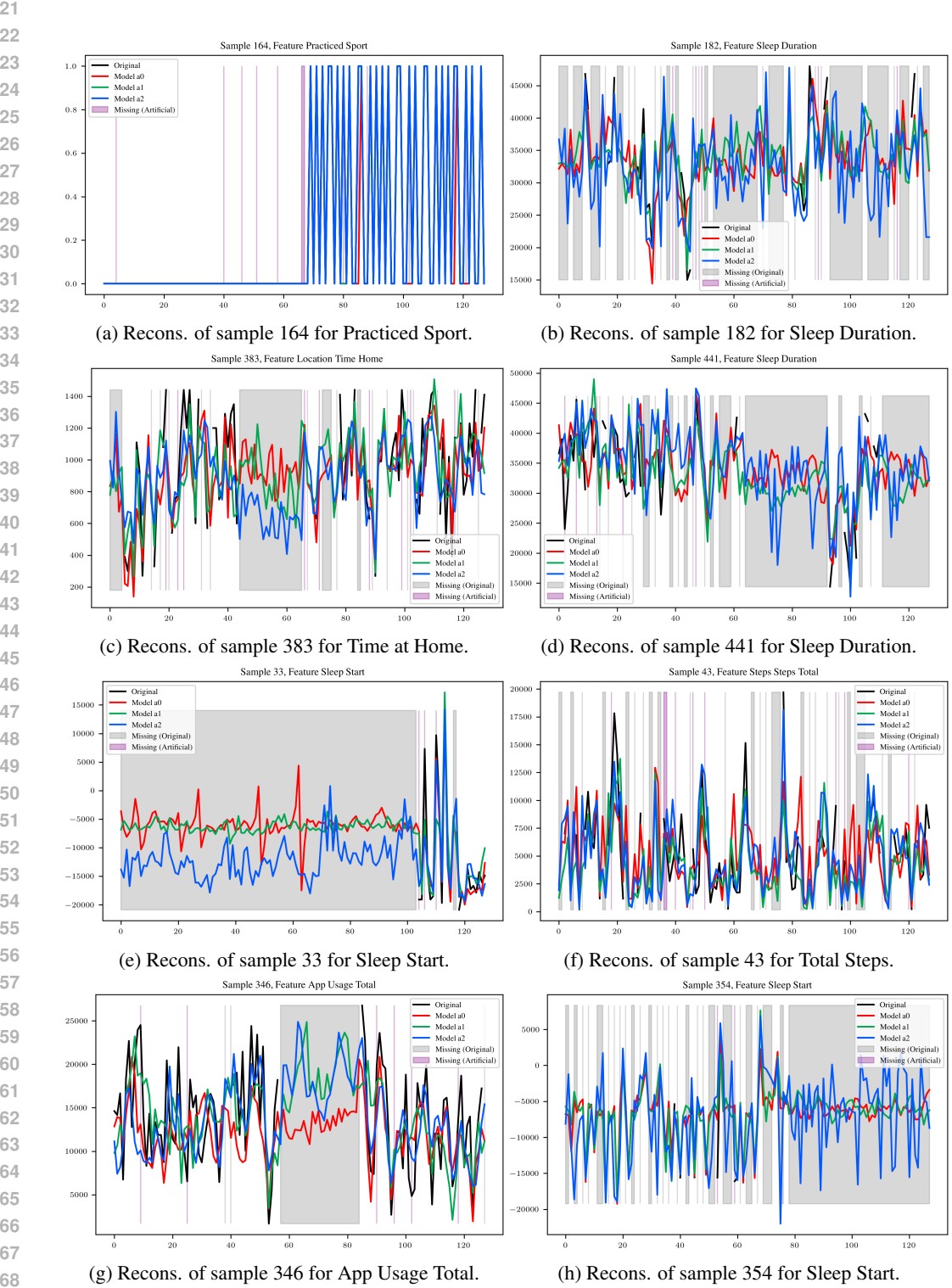

(a) Recons. of sample 164 for Practiced Sport.

(b) Recons. of sample 182 for Sleep Duration.

(c) Recons. of sample 383 for Time at Home.

(d) Recons. of sample 441 for Sleep Duration.

(e) Recons. of sample 33 for Sleep Start.

(f) Recons. of sample 43 for Total Steps.

(g) Recons. of sample 346 for App Usage Total.

(h) Recons. of sample 354 for Sleep Start.

Figure 9: Representative signal reconstructions for observed and imputed instances. In cases where the original signal is not explicitly shown, it is because one or more of the models (whose reconstructions are plotted) overlap the true signal precisely, obscuring the original data.

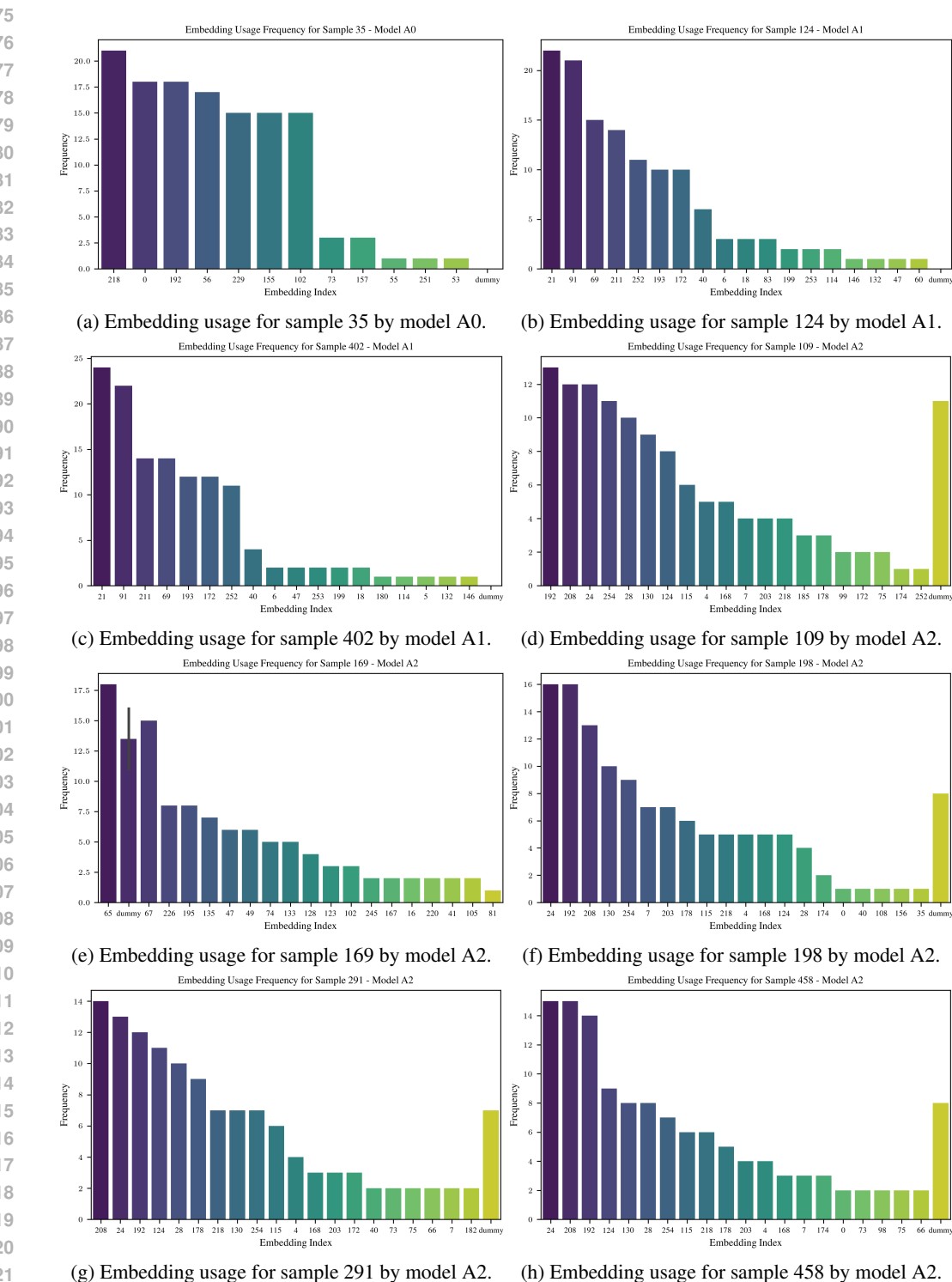

(a) Embedding usage for sample 35 by model A0.

(b) Embedding usage for sample 124 by model A1.

(c) Embedding usage for sample 402 by model A1.

(d) Embedding usage for sample 109 by model A2.

(e) Embedding usage for sample 169 by model A2.

(f) Embedding usage for sample 198 by model A2.

(g) Embedding usage for sample 291 by model A2.

(h) Embedding usage for sample 458 by model A2.

Figure 10: Embedding usage histograms for different samples. Out of the total 256 available embeddings, we observe that only a small subset is typically used, resulting in a sparse and more interpretable solution. Embeddings that are individually uncommon are categorized as belonging to the "dummy" embedding, emphasizing the model's focus on a limited number of relevant embeddings.

