# OpenReview forum: "A Foundation Model for Patient Behavior Monitoring and Suicide Detection"
_ICLR.cc/2025/Conference — Submitted to ICLR 2025_

### Official Review · Reviewer_4fFh · 2024-10-23

**Soundness:** 3
**Presentation:** 3
**Contribution:** 3
**Rating:** 6
**Confidence:** 3

**Summary:**

The paper presents a foundational model for imputing PDP data with missing MNAR values using VQ-VAE. This system aims to encode the time series into discretized embeddings that represent latent profiles, which help capture patient behavior and improve performance in downstream tasks.

The results appear comparable to existing alternatives in the literature. However, the strength of this model lies in its flexibility, as it can be applied to tasks for which it was not explicitly trained. This demonstrates that the generated embeddings capture meaningful representations of the input data, allowing their use across a broader range of objectives without requiring fine-tuning.

**Strengths:**

- **Model Flexibility**: One of the standout features of the proposed model is its flexibility. The embeddings learned are generalizable across different tasks, even those for which the model was not explicitly trained. This demonstrates its robustness and potential for transfer learning.

- **Potential for Interpretability**: The use of latent profile embeddings offers promise for improving interpretability in clinical and behavioral data, an essential feature for real-world healthcare applications. This adds an additional layer of value to the model beyond just predictive accuracy.

- **Description of the method**: The article presents a detailed explanation of the method.

**Weaknesses:**

- **Limited Evaluation on Downstream Tasks**: Although the paper emphasizes the model’s flexibility and its applicability to a variety of downstream tasks, the evaluation is limited to only two tasks. Expanding the analysis to include a broader range of scenarios would have provided stronger evidence of the model's generalizability.

- **Lack of Exploration on Interpretability**: The paper asserts that the learned latent profiles enhance interpretability, yet no concrete evidence or case studies are provided to substantiate this claim. A more thorough investigation, possibly through qualitative analysis or correlations with clinically relevant features, would significantly strengthen the paper's contributions.

- **Omission of Spatio-Temporal Architectures**: While the authors mention that exploring spatio-temporal architectures is left for future work, the choice not to use models such as GNNs, RNNs, or Transformers, which are well-suited for these types of tasks, is not sufficiently justified. This omission limits the model's applicability to more complex time series problems.

- **Clarity in Results Presentation:** The presentation of results could be made clearer. The use of multi-line graphs can be confusing, and this would be better supplemented with numerical information in the main text. Additionally, incorporating a wider range of classification metrics beyond AUC, such as AUPRC, would provide a more comprehensive evaluation of the model's performance.

**Questions:**

- When training the model, synthetic missing values are introduced. Why are these not also used during the calculation of the loss function to optimize their imputation, potentially improving the model’s capacity to handle missing values in the time series?

- What is the class imbalance in the Suicide Detection task? If there is significant imbalance, the AUC score might be misleading. In clinical settings, the AUPRC is often used as an alternative metric for imbalanced data [1]. Perhaps it could be considered for this problem as well.

- The imputation errors in Table 6 seem quite high, although this may be due to the nature of the data. Could the authors provide results from comparison techniques? It would be useful to include at least one method of a common benchmark [2], and possibly show results from methods like linear interpolation and mean interpolation.

- The paper mentions that these profile embeddings could enhance the interpretability of individual behaviors. Have the authors explored this? Were they able to find any correlation between the presence of specific embeddings and labels? If not, could they outline how they would approach this in future work, if it falls outside the scope of this paper?

- Why were more suitable architectures for capturing spatio-temporal representations not used? While it is mentioned that this is left for future work, it’s unclear why models such as GNNs, RNNs, Transformers, or even newer blocks like Mamba were not considered [2] [3].

- Figure 4 is difficult to interpret in terms of the quality of imputations and reconstructions. It would be helpful to complement this with a table, similar to Table 6.


[1] Moor, M., Rieck, B., Horn, M., Jutzeler, C. R., & Borgwardt, K. (2021). Early prediction of sepsis in the ICU using machine learning: a systematic review. Frontiers in medicine, 8, 607952.

[2] Cini, A., Marisca, I., & Alippi, C. (2021). Filling the g_ap_s: Multivariate time series imputation by graph neural networks. arXiv preprint arXiv:2108.00298.

[3] Liu, M., Huang, H., Feng, H., Sun, L., Du, B., & Fu, Y. (2023, April). Pristi: A conditional diffusion framework for spatiotemporal imputation. In 2023 IEEE 39th International Conference on Data Engineering (ICDE) (pp. 1927-1939). IEEE.

---

> ### Author Response · Authors · 2024-11-23
> **Answer to Reviewer 3fFh (1/3)**
>
> We would like to thank the reviewer for his comments about the paper and for highlighting the strengths of it. Below, we address the identified weaknesses and respond to the specific questions raised. Note that, as some question were related to some highlighted weaknesses, we will address those already in the weaknesses section.
>
> ## Weaknesses
>
> -   **Limited Evaluation on Downstream Tasks**
>
> We thank the reviewer for his suggestion on this regard. According to this comment, and some of the comments raised by the other reviewers, we have now done the following:
>
> We have included results using the method proposed in the paper: _Arbaizar LP, Lopez-Castroman J, Artés-Rodríguez A, Olmos PM, Ramírez D, “Emotion Forecasting: A Transformer-Based Approach,”_ JMIR Preprints, DOI: 10.2196/preprints.63962, [available here](https://preprints.jmir.org/preprint/63962). In this work, the authors used an imputation method based on hidden Markov models (HMM) for handling missing data and a Transformer deep neural network for time-series forecasting over the same database. Further details on the architecture can be found in their paper.
>
> To compare performance, we **applied the HMM-Transformer method by Arbaizar et al. to the suicide detection downstream tasks** presented in our study. With the VQ-VAE, we leveraged its discrete and sparse latent representations, enabling us to project data into either a single integer (indicating the closest profile) or a vector of pseudo-probabilities of a fixed length for the most relevant profiles. In contrast, the HMM-Transformer model produces a real-valued latent space with no inherent interpretability, employing embeddings of 64 dimensions with values ranging from −1 to 1. The CPD algorithm can deal with all three input types (categorical profile, probabilistic profile or real values). However, when processing the high-dimensional real-valued embeddings produced by the HMM-Transformer, the run length collapsed due to very low predictive posterior probabilities (current observation is always considered to be distinct from past data and therefore changes are prompted at all points). This behavior occurs due to the inability of the CPD to process high dimensional variables, and it evidences the need for latent representations that are low-dimensional and easy to manage, such as those returned by the VQ-VAE.
>
> In addition, we expanded our analysis by introducing a **new downstream task**: **sentiment analysis**, which was also performed in the paper by Arbaizar et al. In this new experiment, we excluded 758 subjects from our database during the VQ-VAE training phase. These individuals, during their respective cohort studies, logged a total of 68,219 emotions alongside daily habits recorded via passive digital phenotyping. Emotion valence—classified as 0 (negative), 1 (neutral), or 2 (positive)—was set as the target variable, predicted using all other passive variables. Predictions were made using an XGBoost classifier.
>
> The results of this task revealed interesting insights (see [https://figshare.com/s/402cab5336581f6a6fb9](https://figshare.com/s/402cab5336581f6a6fb9)). Using the same training and test partitions, the Transformer-based approach achieved a remarkable AUC score of 0.98, while the VQ-VAE also performed well, achieving an AUC of 0.81. This result is noteworthy considering that the VQ-VAE model was not explicitly designed or fine-tuned for this task, making it applicable to new patients without additional adjustments. In contrast, Arbaizar et al.’s architecture required three separate stages—each requiring training—to impute missing data (HMM), forecast the next sequence (Transformer), and classify emotions (XGBoost).
>
> The VQ-VAE offers a distinct advantage by combining the first two steps into a unified approach. It can project sequences with missing data directly into latent embeddings without additional intervention for imputation or preparation, demonstrating its versatility and efficiency compared to task-specific architectures.

---

> > ### Author Response · Authors · 2024-11-23
> > **Answer to Reviewer 3fFh (2/3)**
> >
> > -   **Lack of Exploration on Interpretability**
> >
> > This comment was also raised by other reviewers. We have worked on providing specific analyses supporting our claim. In particular:
> >
> > We have conducted an exploratory analysis leveraging the discrete latent representations learned by the VQ-VAE model, applying Latent Dirichlet Allocation (LDA) to identify meaningful groupings of mental health patients based on the behavioral covariates used during training. Figures related to this analysis can be found at the following link: [https://figshare.com/s/3b97194103b7b5b6254c](https://figshare.com/s/3b97194103b7b5b6254c).
> >
> > This analysis uncovered distinct topics, corresponding to clusters of patients with shared behavioral patterns. The discrete embeddings, which map directly to the original variables, enabled us to interpret these topics in clinically relevant terms. For example, LDA identified four prominent topics, each represented by the top five most relevant embeddings decoded into original behavioral features. These embeddings capture different types of days, reflecting variations in patient behavior over time.
> >
> > Key findings include the following:
> >
> > -   Topics 0 and 2 represent more active patients with higher mobility, while Topics 1 and 3 correspond to less active, more stationary patients.
> > -   Topics 1 and 3 share two embeddings among their top five features, indicating overlapping behavioral characteristics.
> > -   A significant distinction emerged between Topics 0 and 2: Topic 2 demonstrates a more stable distribution of embeddings, suggesting consistent patterns in behavioral metrics such as location distance, sleep start times, sleep duration, and time spent at home. Conversely, Topic 0 displays greater variability across these metrics, reflecting a broader range of behavioral patterns.
> >
> > This variability may hold clinical significance, potentially indicating differences in behavioral regularity or flexibility, which are relevant factors for assessing patient well-being. To ensure meaningful comparisons, all feature values were normalized prior to analysis.
> >
> > These findings demonstrate the capability of our approach to uncover nuanced patient characteristics and variability through the latent representations, enhancing the interpretability and potential clinical utility of the model. We will include this analysis, along with visualizations of the identified topics, in the final manuscript to further highlight the applicability of the proposed method to real-world clinical decision-making.
> >
> > -   **Omission of Spatio-Temporal Architectures**
> >
> > As mentioned in the response to the first comment, we have included results using the method proposed in the paper: _Arbaizar LP, Lopez-Castroman J, Artés-Rodríguez A, Olmos PM, Ramírez D, “Emotion Forecasting: A Transformer-Based Approach,”_ JMIR Preprints, DOI: 10.2196/preprints.63962, [available here](https://preprints.jmir.org/preprint/63962). In this work, the authors used an imputation method based on hidden Markov models (HMM) for handling missing data and a Transformer deep neural network for time-series forecasting over the same database. Further details on the architecture can be found in their paper.
> >
> > However, due to time constraints, we have not yet tested additional models, particularly GNNs, which we agree would be an interesting avenue for future exploration in this type of problem.

---

> > > ### Author Response · Authors · 2024-11-23
> > > **Answer to Reviewer 3fFh (3/3)**
> > >
> > > -   **Clarity in Results Presentation**
> > >
> > > Thank you for highlighting the importance of introducing additional scores, such as the AUPRC. Our dataset (and any dataset that may work in the context of suicidal behavior) is extremely unbalanced because only a few clinical suicide attempts happen among several thousand days. In our case, 14 events were reported for a whole set of over 90,000 registered days, considering all patients. In similar cases, the use of AUPRC is recommended. However, we attempted to implement precision-recall curves as suggested but, due to the imbalance mentioned, precision values are extremely low because there are much more false alarms than true positives. While false alarms should ideally be minimized, they are to some extent acceptable in the context of suicide prevention, where the focus is on effectively identifying and treating all patients with a risk of attempting suicide. Furthermore, these false alarms may be signs of instability or changes in the subject’s life, and dedicating special attention to the patient during these moments may be of clinical importance and improve the quality of mental health monitoring.
> > >
> > > [[https://figshare.com/s/90ce6ddf6dc3598fd5df](https://figshare.com/s/90ce6ddf6dc3598fd5df)]
> > >
> > > We provide a figure showing an example of both ROC and precision-recall curves for the same data, using several values of λ. It can be seen that the best AUPRC values lie in the order of 10−4, and that the curves are not easily interpreted, since they lie in a very low and narrow range of precision values. In contrast, ROC curves seem to provide more significance on the results, even if the model’s low precision is not accurately informed. Still, because of the reasons mentioned above, we believe that such precision may not be so relevant and that the AUROC is more suitable in this particular case.
> > >
> > > ## Questions
> > >
> > > -   **When training the model, synthetic missing values are introduced. Why are these not also used during the calculation of the loss function to optimize their imputation, potentially improving the model’s capacity to handle missing values in the time series?**
> > >
> > > We thank the reviewer for their insightful question. This is a point we carefully considered in our previous discussions and double-checked during our analysis, finding no significant differences. To assess the robustness of our approach, we introduced synthetic missing values to evaluate the performance of our reconstruction method. However, training on synthetic missing data was not our primary objective, as our goal is to predict missing values based on the observed time-series data. While we agree that this is a valid training approach, our experiments revealed no significant improvements. Therefore, we opted to retain the original methodology.
> > >
> > > -   **Figure 4 is difficult to interpret in terms of the quality of imputations and reconstructions. It would be helpful to complement this with a table, similar to Table 6.**
> > >
> > > We appreciate the reviewer’s observation regarding the interpretability of Figure 4. While we understand that it may be challenging to interpret in isolation, we included it as a representative visualization of the data imputation quality because it is more concise and easier to present within the main body of the paper. Due to space limitations, detailed tables were placed in the supplementary material, although we agree it would have been ideal to include both the visualization and the tables in the main text for a more comprehensive presentation.

---

> > > > ### Comment · Reviewer_4fFh · 2024-11-23
> > > > **Response to rebuttal**
> > > >
> > > > The authors have addressed all my questions, and although they were not always able to add more content to the paper, they provided well-detailed responses with clear reasoning behind their decisions. Therefore, I do not consider the lack of additional content to be an issue per se. I would like to thank the authors for thoroughly addressing my main concerns and for their dedication in providing such robust answers. Given this, I have decided to increase my initial score from 5 to 6.

---

### Official Review · Reviewer_PvXG · 2024-10-24

**Soundness:** 2
**Presentation:** 2
**Contribution:** 2
**Rating:** 3
**Confidence:** 4

**Summary:**

This paper explores patient behavior monitoring and suicide detection by adapting the vector quantized variational autoencoder (VQ-VAE) to analyze data from real-world wearable devices. By reconstructing heterogeneous multi-source time-series data and accounting for patterns of missing data, the proposed model is validated to be effective in detecting change-time events within a cohort of suicidal patients.

**Strengths:**

S1. This research provides significant medical potential by modeling heterogeneous, multi-source time-series data from wearable devices, aiming to develop advanced approaches for identifying potential behavioral shifts that signal serious mental health risks.

S2. The proposed model, which leverages VQ-VAE as the foundation model, captures the missing data patterns, and incorporates a change-point detection algorithm, is overall reasonable from a technical standpoint for this specific medical application.

S3. The experimental evaluation includes a demonstration of imputation performance, an assessment of downstream tasks, results from statistical tests, and several representative case studies.

**Weaknesses:**

W1. My primary concern regarding this paper is its technical novelty. Although the proposal adeptly uses the VQ-VAE model as the foundational framework, modifies its design to capture missing data patterns in time-series wearable device data, and connects to change-point detection as a downstream task, it relies heavily on existing techniques. This reliance diminishes the perceived novelty of the proposed model. Moreover, the paper does not sufficiently articulate the challenges and complexities associated with integrating these techniques, which is crucial for demonstrating its innovation.

W2. The evaluation of imputation performance lacks comparative baselines, which undermines the robustness of the findings. Additionally, the downstream task evaluation includes only one baseline, HetMM, which is insufficient. Incorporating a broader range of advanced and recent baselines would help demonstrate the superiority of the proposal in this research area.

W3. The evaluation relies solely on a single proprietary dataset, limiting the generalizability of the findings. Expanding the evaluation to include multiple datasets would enhance the credibility and applicability of the proposed model across different contexts.

W4. The performance improvements offered by the proposed model are marginal when compared with HetMM. Given that the proposal claims advantages in scalability and efficiency, it is essential to substantiate these claims with experiments on larger datasets or real-time analysis scenarios.

W5. As the proposal targets a specific healthcare application—suicide detection—it would be advantageous to include medical validation of the findings or to offer insights into how the proposed model could inform real-world clinical decision-making. Such contributions would significantly elevate the practical relevance of the proposal.

W6. The writing of this paper could be improved by addressing the following aspects:

(i) The paper should clearly identify which model among Model A0, Model A1, and Model A2 performs best and should be recognized as the final proposed model.

(ii) The inclusion of a detailed related work section is necessary. This section would help situate the paper within the existing body of knowledge, providing a comprehensive backdrop against which the current research can be assessed and appreciated.

**Questions:**

Please refer to points W1 through W6 for detailed concerns and recommendations.

---

> ### Author Response · Authors · 2024-11-23
> **Answer to Reviewer PvXG (1/4)**
>
> We would like to thank the reviewer for his comments about the paper and for highlighting the strengths of it. Below, we address the identified weaknesses and respond to the specific questions raised.
>
> ## Weaknesses
>
> -   **Technical novelty**
>
> As this comment was also raised by the previous reviewers, we next copy the response used to address their concerns and explain how the novelty of our paper should be understood from our point of view:
>
> First, we would like to clarify that our paper is not intended as a “method paper.” We are not proposing a novel architecture; instead, we identify VQ-VAE as an effective choice among state-of-the-art models for data imputation in heterogeneous real-world datasets with substantial missing data.
>
> Additionally, we emphasize the unique advantages of VQ-VAE for our specific application. Its discrete latent space enables downstream tasks without any fine-tuning such as suicide detection and sentiment analysis—an extension for which we already have promising results, to be included in the revised submission, and shown in the figures linked to the previous comment. Our work demonstrates that within the existing state-of-the-art, certain architectures, like VQ-VAE, are particularly well-suited for addressing the challenges posed by highly complex health time-series data recorded from wearable and mobile devices. This includes managing significant missingness and heterogeneity while facilitating solutions for diverse downstream tasks without requiring explicit training on specific patient subpopulations.
>
> The defining feature of VQ-VAE is its discrete latent space, which enables competitive performance in the two downstream tasks we evaluated. By contrast, the HMM-Transformer we included for comparison performs well in downstream prediction tasks such as sentiment analysis, for which it was designed, but struggles with suicide detection. This limitation stems from its reliance on a high-dimensional continuous latent space, where statistical changes in behavioral patterns are obscured. The complex, continuous input it provides to the CPD algorithm makes it less effective for detecting subtle shifts necessary for this application.
>
> -   **The performance improvements offered by the proposed model are marginal when compared with HetMM. Given that the proposal claims advantages in scalability and efficiency, it is essential to substantiate these claims with experiments on larger datasets or real-time analysis scenarios.**
>
> We have not included experiments specifically to demonstrate the scalability of our solution because the improved scalability of VQ-VAE is inherent to its design. Unlike HetMM, which requires a separate model for each patient, our VQ-VAE approach leverages a pre-trained model that can reconstruct signals with missing data without the need for user-specific fine-tuning. For downstream tasks, we simply take the pre-trained model, encode the data from a new user whose data was not seen during training, and use those encodings in the VQ-VAE dictionary as input for the downstream task algorithm. This means that generating the user profile, which serves as input for the downstream task algorithm, is independent of the number of users for whom we want to perform the task. As a result, this approach is inherently more scalable and efficient than HetMM, which requires a separate model for each patient.

---

> > ### Author Response · Authors · 2024-11-23
> > **Answer to Reviewer PvXG (2/4)**
> >
> > -   **Imputation performance lacks comparative baselines. The downstream task evaluation includes only one baseline, HetMM, which is insufficient. Incorporating a broader range of advanced and recent baselines would help demonstrate the superiority of the proposal in this research area.**
> >
> > We thank the reviewer for his suggestion on this regard. According to this comment, and some of the comments raised by the other reviewers, we have now done the following:
> >
> > We have included results using the method proposed in the paper: _Arbaizar LP, Lopez-Castroman J, Artés-Rodríguez A, Olmos PM, Ramírez D, “Emotion Forecasting: A Transformer-Based Approach,”_ JMIR Preprints, DOI: 10.2196/preprints.63962, [available here](https://preprints.jmir.org/preprint/63962). In this work, the authors used an imputation method based on hidden Markov models (HMM) for handling missing data and a Transformer deep neural network for time-series forecasting over the same database. Further details on the architecture can be found in their paper.
> >
> > To compare performance, we **applied the HMM-Transformer method by Arbaizar et al. to the suicide detection downstream tasks** presented in our study. With the VQ-VAE, we leveraged its discrete and sparse latent representations, enabling us to project data into either a single integer (indicating the closest profile) or a vector of pseudo-probabilities of a fixed length for the most relevant profiles. In contrast, the HMM-Transformer model produces a real-valued latent space with no inherent interpretability, employing embeddings of 64 dimensions with values ranging from −1 to 1. The CPD algorithm can deal with all three input types (categorical profile, probabilistic profile or real values). However, when processing the high-dimensional real-valued embeddings produced by the HMM-Transformer, the run length collapsed due to very low predictive posterior probabilities (current observation is always considered to be distinct from past data and therefore changes are prompted at all points). This behavior occurs due to the inability of the CPD to process high dimensional variables, and it evidences the need for latent representations that are low-dimensional and easy to manage, such as those returned by the VQ-VAE.
> >
> > In addition, we expanded our analysis by introducing a **new downstream task**: **sentiment analysis**, which was also performed in the paper by Arbaizar et al. In this new experiment, we excluded 758 subjects from our database during the VQ-VAE training phase. These individuals, during their respective cohort studies, logged a total of 68,219 emotions alongside daily habits recorded via passive digital phenotyping. Emotion valence—classified as 0 (negative), 1 (neutral), or 2 (positive)—was set as the target variable, predicted using all other passive variables. Predictions were made using an XGBoost classifier.
> >
> > The results of this task revealed interesting insights (see [https://figshare.com/s/402cab5336581f6a6fb9](https://figshare.com/s/402cab5336581f6a6fb9)). Using the same training and test partitions, the Transformer-based approach achieved a remarkable AUC score of 0.98, while the VQ-VAE also performed well, achieving an AUC of 0.81. This result is noteworthy considering that the VQ-VAE model was not explicitly designed or fine-tuned for this task, making it applicable to new patients without additional adjustments. In contrast, Arbaizar et al.’s architecture required three separate stages—each requiring training—to impute missing data (HMM), forecast the next sequence (Transformer), and classify emotions (XGBoost).
> >
> > The VQ-VAE offers a distinct advantage by combining the first two steps into a unified approach. It can project sequences with missing data directly into latent embeddings without additional intervention for imputation or preparation, demonstrating its versatility and efficiency compared to task-specific architectures.
> >
> > -   **Private dataset**
> >
> > We agree that working with private datasets can hinder the reproducibility of results and limit progress in this research area. To address this, if our paper is accepted, we commit to publicly releasing at least a partial dataset that has already been disclosed.

---

> > > ### Author Response · Authors · 2024-11-23
> > > **Answer to Reviewer PvXG (3/4)**
> > >
> > > -   **As the proposal targets a specific healthcare application—suicide detection—it would be advantageous to include medical validation of the findings or to offer insights into how the proposed model could inform real-world clinical decision-making. Such contributions would significantly elevate the practical relevance of the proposal.**
> > >
> > > We appreciate the reviewer’s interest in exploring how the proposed model can inform real-world clinical decision-making beyond the presented downstream applications, such as suicide detection via CPD. To address this, we have conducted an exploratory analysis leveraging the discrete latent representations learned by the VQ-VAE model, applying Latent Dirichlet Allocation (LDA) to identify meaningful groupings of mental health patients based on the behavioral covariates used during training. Figures related to this analysis can be found at the following link: [https://figshare.com/s/3b97194103b7b5b6254c](https://figshare.com/s/3b97194103b7b5b6254c).
> > >
> > > This analysis uncovered distinct topics, corresponding to clusters of patients with shared behavioral patterns. The discrete embeddings, which map directly to the original variables, enabled us to interpret these topics in clinically relevant terms. For example, LDA identified four prominent topics, each represented by the top five most relevant embeddings decoded into original behavioral features. These embeddings capture different types of days, reflecting variations in patient behavior over time.
> > >
> > > Key findings include the following:
> > >
> > > -   Topics 0 and 2 represent more active patients with higher mobility, while Topics 1 and 3 correspond to less active, more stationary patients.
> > > -   Topics 1 and 3 share two embeddings among their top five features, indicating overlapping behavioral characteristics.
> > > -   A significant distinction emerged between Topics 0 and 2: Topic 2 demonstrates a more stable distribution of embeddings, suggesting consistent patterns in behavioral metrics such as location distance, sleep start times, sleep duration, and time spent at home. Conversely, Topic 0 displays greater variability across these metrics, reflecting a broader range of behavioral patterns.
> > >
> > > This variability may hold clinical significance, potentially indicating differences in behavioral regularity or flexibility, which are relevant factors for assessing patient well-being. To ensure meaningful comparisons, all feature values were normalized prior to analysis.
> > >
> > > These findings demonstrate the capability of our approach to uncover nuanced patient characteristics and variability through the latent representations, enhancing the interpretability and potential clinical utility of the model. We will include this analysis, along with visualizations of the identified topics, in the final manuscript to further highlight the applicability of the proposed method to real-world clinical decision-making.
> > >
> > > -   **Discussion on Models A0, A1 and A2 variants**
> > >
> > > Since another reviewer also raised this specific issue, here is the response about it and how we will address the concern in the final version of the paper:
> > >
> > > We sincerely thank the reviewer for raising this important point. In our initial submission, we did not emphasize a definitive conclusion comparing the performance of the VQ-VAE variants due to their similar and comparable results, as shown in Figures 4 and 8, as well as Tables 6, 8, and 9. However, upon closer examination, it becomes clear that the A0 variant achieves performance comparable to—and in some cases exceeding—that of the missingness-mask-conditioned variants. This highlights the robustness of A0 as a simple yet effective model for the diverse tasks analyzed in the paper.
> > >
> > > Models A1 and A2, which explicitly encode the missingness mask, primarily serve to validate that Model A0, which implicitly incorporates the missingness mask through the zero-imputation mechanism, is sufficiently robust.
> > >
> > > We appreciate the opportunity to clarify this aspect and will incorporate this discussion into the revised manuscript.

---

> > > > ### Author Response · Authors · 2024-11-23
> > > > **Answer to PvXG (4/4)**
> > > >
> > > > -   **A related work section is necessary**
> > > >
> > > > In the final version of the manuscript, we will include a dedicated section on related work, incorporating additional findings inspired by the reviewer’s suggestions.
> > > >
> > > > This section will explicitly review other models for time-series forecasting, reconstruction, and missing data imputation. However, we would like to emphasize that what distinguishes our work is the use of a discrete latent space within the VQ-VAE framework. To the best of our knowledge, no prior studies have employed a discrete latent-space model for reconstructing heterogeneous missing data in time series derived from wearable and mobile devices. This distinctive approach is specifically tailored to the unique characteristics of our dataset and the downstream tasks explored in this study.
> > > >
> > > > For example, a recent study integrating state-of-the-art techniques for time-series prediction is:
> > > >
> > > > **Moreno-Pino, F., Arroyo, Á., Waldon, H., Dong, X., & Cartea, Á. (2024). Rough Transformers: Lightweight Continuous-Time Sequence Modelling with Path Signatures. _arXiv preprint_ arXiv:2405.20799.**
> > > >
> > > > This paper introduces the Rough Transformer, a variation of the Transformer model that operates on continuous-time representations of input sequences while achieving significantly lower computational costs. However, it does not employ a discrete latent space capable of supporting the type of downstream tasks demonstrated in our work.
> > > >
> > > > Another relevant study that we reference in the updated manuscript is:
> > > >
> > > > **Fortuin, V., Hüser, M., Locatello, F., Strathmann, H., & Rätsch, G. (2018). Som-VAE: Interpretable Discrete Representation Learning on Time Series. _arXiv preprint_ arXiv:1806.02199.**
> > > >
> > > > In this study, the authors address the challenge of non-differentiability in discrete latent spaces similarly to VQ-VAE and introduce a Markov model in the representation space to enhance interpretability. Their evaluation includes static (Fashion-)MNIST data, time-series data derived from linearly interpolated (Fashion-)MNIST images, a chaotic Lorenz attractor system with two macro-states, and a medical time-series application using the eICU dataset.
> > > >
> > > > It is worth noting that the eICU dataset differs fundamentally from our own. It consists of lab measurements aggregated from multiple critical care units across the United States, whereas our dataset is derived from wearable and mobile devices. Moreover, none of the datasets used in these studies exhibit the high heterogeneity or structured missing data patterns characteristic of our work. These challenges underscore the relevance and novelty of our approach in addressing complex, real-world time-series data.

---

> > > > > ### Comment · Reviewer_PvXG · 2024-11-23
> > > > > **Response to rebuttal**
> > > > >
> > > > > I appreciate the authors' detailed rebuttal, particularly the inclusion of the HMM-Transformer baseline, the evaluation of an additional downstream task, and the LDA-based exploratory analysis.
> > > > > However, based on the clarifications provided regarding the paper's novelty and intended scope, I believe that investigating an existing method within a specific application scenario may not strongly appeal to the readership of ICLR. Therefore, I remain concerned about the paper's novelty and will maintain my current rating.

---

> > > > > > ### Author Response · Authors · 2024-11-26
> > > > > >
> > > > > > Thank you for your prompt response.
> > > > > >
> > > > > > We understand and respect your position regarding the paper's novelty due to the use of an existing method. However, we would like to point out that our submission was made to the "Applications to Neuroscience & Cognitive Science" area, where the focus, as we understand, is more on the application of methods rather than on methodological innovations. While we fully appreciate your perspective, we wanted to raise this clarification in the hope that it might offer an alternative lens through which to view our paper, given the scope of this ICLR 2025 area.

---

### Official Review · Reviewer_W2p2 · 2024-11-03

**Soundness:** 3
**Presentation:** 3
**Contribution:** 3
**Rating:** 6
**Confidence:** 3

**Summary:**

The paper introduces a foundation model for patient behavior monitoring and suicide detection based on a modified Vector Quantized Variational Autoencoder (VQ-VAE). This model effectively reconstructs heterogeneous time-series data from wearable devices and captures missing data patterns. The paper highlights its application in predicting suicide attempts using a probabilistic change-point detection (CPD) algorithm on data from a cohort of psychiatric patients. The model shows promise for scalable, efficient patient monitoring by outperforming patient-specific methods without requiring fine-tuning.

**Strengths:**

The approach is novel in integrating VQ-VAE with CPD for behavioral analysis and suicide prediction, particularly in using discrete latent representations for temporal health data. The paper is methodologically strong, with well-designed experiments and performance evaluations. The work has substantial potential in clinical monitoring and predictive modeling, addressing a critical healthcare application with an approach that could be extended to other domains. The paper is well-written, with clear articulation of concepts and objectives, making complex methodologies accessible. Figures and tables are effectively used to illustrate the architecture and experimental results.

**Weaknesses:**

The evaluation is conducted on a single private dataset.
The use of a proprietary dataset limits reproducibility and prevents validation or benchmarking with public datasets, reducing transparency. Expanding to public datasets could provide a more comprehensive assessment of the model's generalization and facilitate future comparisons.

The study lacks a comparison with other generative models for reconstruction tasks, which could demonstrate the superiority or limitations of the proposed VQ-VAE model, comparisons with more generative models in the reconstruction phase could bolster the paper’s robustness.

Although three VQ-VAE variants (A0, A1, and A2) are tested, there is limited discussion on the specific performance differences across these variants and their impact on handling missing data.

Certain technical details, such as the choice of hyperparameters, could be clarified to enhance reproducibility.

**Questions:**

1. Could you elaborate on the performance differences among the three VQ-VAE variants (A0, A1, and A2) in terms of handling specific missing data patterns?

2. Have you considered evaluating the model on additional, publicly available behavioral datasets to better assess generalizability and facilitate reproducibility? Do the authors plan to publish or share the dataset to reproduce and enable further research validation?

3. Have you explored using alternative generative models, such as GANs or VAEs, for comparison on data reconstruction? How do you envision these models performing relative to VQ-VAE in terms of capturing behavioral patterns?

4. Can the authors clarify the rationale for choosing specific hyperparameters for CPD (e.g., the hazard function parameter λ)?

A minor typo: Line 188 - The term "D" is used without prior definition or context.

---

> ### Author Response · Authors · 2024-11-22
> **Answer to Reviewer W2p2 (1/2)**
>
> First, we sincerely thank the reviewer for their insightful comments on the paper. Below, we address the identified weaknesses and respond to the specific questions raised. Note that, as some question were related to some highlighted weaknesses, we will address those already in the weaknesses section.
>
> ### Weaknesses
>
> -   **Private dataset**
>
> We agree that working with private datasets can hinder the reproducibility of results and limit progress in this research area. To address this, if our paper is accepted, we commit to publicly releasing at least a partial dataset that has already been disclosed.
>
> -   **Comparison with other generative models for reconstruction tasks**
>
> This comment was also raised by Reviewer 1 (Sgoo). For this reason, we include here the same answer provided to that reviewer, hoping that it serves to address the concern of Reviewer 2 (W2p2):
>
> We acknowledge that our initial submission lacked this critical analysis, and we have addressed this by including results using the method proposed in the paper: _Arbaizar LP, Lopez-Castroman J, Artés-Rodríguez A, Olmos PM, Ramírez D, “Emotion Forecasting: A Transformer-Based Approach,”_ JMIR Preprints, DOI: 10.2196/preprints.63962, [available here](https://preprints.jmir.org/preprint/63962). In this work, the authors used an imputation method based on hidden Markov models (HMM) for handling missing data and a Transformer deep neural network for time-series forecasting over the same database. Further details on the architecture can be found in their paper.
>
> To compare performance, we **applied the HMM-Transformer method by Arbaizar et al. to the suicide detection downstream tasks** presented in our study. With the VQ-VAE, we leveraged its discrete and sparse latent representations, enabling us to project data into either a single integer (indicating the closest profile) or a vector of pseudo-probabilities of a fixed length for the most relevant profiles. In contrast, the HMM-Transformer model produces a real-valued latent space with no inherent interpretability, employing embeddings of 64 dimensions with values ranging from −1 to 1.
>
> In addition, we expanded our analysis by introducing a **new downstream task**: **sentiment analysis**, which was also performed in the paper by Arbaizar et al. In this new experiment, we excluded 758 subjects from our database during the VQ-VAE training phase. These individuals, during their respective cohort studies, logged a total of 68,219 emotions alongside daily habits recorded via passive digital phenotyping. Emotion valence—classified as 0 (negative), 1 (neutral), or 2 (positive)—was set as the target variable, predicted using all other passive variables. Predictions were made using an XGBoost classifier.
>
> The results of this task revealed interesting insights (see [https://figshare.com/s/402cab5336581f6a6fb9](https://figshare.com/s/402cab5336581f6a6fb9)). Using the same training and test partitions, the Transformer-based approach achieved a remarkable AUC score of 0.98, while the VQ-VAE also performed well, achieving an AUC of 0.81. This result is noteworthy considering that the VQ-VAE model was not explicitly designed or fine-tuned for this task, making it applicable to new patients without additional adjustments. In contrast, Arbaizar et al.’s architecture required three separate stages—each requiring training—to impute missing data (HMM), forecast the next sequence (Transformer), and classify emotions (XGBoost).
>
> The VQ-VAE offers a distinct advantage by combining the first two steps into a unified approach. It can project sequences with missing data directly into latent embeddings without additional intervention for imputation or preparation, demonstrating its versatility and efficiency compared to task-specific architectures.
>
> -   **Discussion on Models A0, A1 and A2 variants**
>
> We sincerely thank the reviewer for raising this important point. In our initial submission, we did not emphasize a definitive conclusion comparing the performance of the VQ-VAE variants due to their similar and comparable results, as shown in Figures 4 and 8, as well as Tables 6, 8, and 9. However, upon closer examination, it becomes clear that the A0 variant achieves performance comparable to—and in some cases exceeding—that of the missingness-mask-conditioned variants. This highlights the robustness of A0 as a simple yet effective model for the diverse tasks analyzed in the paper.
>
> Models A1 and A2, which explicitly encode the missingness mask, primarily serve to validate that Model A0, which implicitly incorporates the missingness mask through the zero-imputation mechanism, is sufficiently robust.
>
> We appreciate the opportunity to clarify this aspect and will incorporate this discussion into the revised manuscript.

---

> > ### Author Response · Authors · 2024-11-22
> > **Answer to Reviewer W2p2 (2/2)**
> >
> > ### Questions
> >
> > -   **Certain technical details, such as the choice of hyperparameters, could be clarified to enhance reproducibility.**
> >
> > The discussion on hyperparameters was deferred to the supplementary material due to space limitations. However, in the revised version of the manuscript, which will be reformatted to incorporate the insightful feedback from this discussion, we will include additional key details about the hyperparameters used in our VQ-VAE model. This will provide greater clarity and ease of understanding for potential readers.
> >
> > -   **Can the authors clarify the rationale for choosing specific hyperparameters for CPD (e.g., the hazard function parameter λ)?**
> >
> > In the case of the hazard function parameter λ, because its choice greatly affects the performance of the model, we decided to provide three different values in the ROC curves to allow for a richer comparison between the models. Still, there is no interpretability on the exact value of this parameter and the ultimate value would depend on the model to be used and application of interest). During our experiments, a broader grid search was conducted with λ ranging from 10^−20 to 10^19, and the best ones overall were included in the submitted paper. Only three values were chosen to facilitate cleaner plots that are easier to analyze.
> >
> > Number of profiles (_K_) and size of the temporal window (_w_): these two hyperparameters are positive integers and should not take large values (both to avoid computational cost to increase and because it is not interesting from a practical point of view). A few values were tested and the best ones were chosen for the final experiments (see Figure 6 in Appendix D of the submitted manuscript).
> >
> > The value of hyperparameter _S_, which indicates the number of samples drawn from the multinomial distribution, was chosen based on our previous experience working with this type of model. Other CPD configurations not mentioned in the submitted paper were also set according to insights gathered from past research.
> >
> > -   **A minor typo: Line 188 - The term "D" is used without prior definition or context.**
> >
> > We thank the reviewer for catching this undefined variable. We will add the corresponding definition to the final version of the manuscript.

---

> ### Comment · Reviewer_W2p2 · 2024-11-24
> **Response to rebuttal**
>
> Thank you for the detailed and thoughtful responses to the raised concerns. While your responses effectively address many of the points raised, additional details on the planned dataset release and broader comparisons with other generative models could further strengthen the paper. I will maintain my current rating for this paper and appreciate your effort in addressing the feedback.

---

### Official Review · Reviewer_Sgoo · 2024-11-04

**Soundness:** 2
**Presentation:** 3
**Contribution:** 1
**Rating:** 3
**Confidence:** 3

**Summary:**

The authors introduce a model based on VQ-VAE, trained in a self-supervised way with different (informative) ways of modeling missingness. The authors look at reconstruction and imputation accuracy of the model on a behavioral data set of wearable readings. Using the trained model the authors implement a Bayesian online learning procedure to infer the MAP of the presence of a change point in the data with proposed application to suicide prevention, which is benchmarked against HetMM on their behavioral data set in terms of ROC curves.

**Strengths:**

- Time series modeling and change point detection are important problems in clinical applications

**Weaknesses:**

- A related work section for other time series models is missing, but important to place the work within the context of the broader literature
- For imputation, the model should be compared to baselines.
- Other baselines for change-point detection than HetMM would be beneficial, e.g. point process frameworks, time series foundation models
- The novelty of the work (as opposed to applying a known framework to a novel interesting problem) is not entirely clear to me.

**Questions:**

- What is the calibration of the pseudo-probabilities as compared to HetMM?
- How does the model deal with censoring due to loss of follow-up?

---

> ### Author Response · Authors · 2024-11-22
> **Answer to Reviewer Sgoo (1/2)**
>
> We would like to thank the reviewer for their insightful comments and constructive feedback. We kindly invite the reviewer to carefully read our responses, where we have addressed their suggestions to improve the paper and strive to meet the acceptance threshold. Below, we provide our detailed responses to the identified weaknesses and questions:
>
> ### Weaknesses
>
> -   **A related work section for other time series models is missing**
>
> In the updated version of the manuscript, we will include an explicit related work section reviewing other models for time-series forecasting, reconstruction, and missing data imputation. However, we want to highlight that what sets our work apart is the discrete latent space employed by the VQ-VAE. To the best of our knowledge, no previous studies have used this type of discrete latent-space model for reconstructing heterogeneous missing data in time series from wearable and mobile devices. This unique approach is specifically motivated by the characteristics of the dataset and the downstream tasks addressed in this work.
>
> For example, a recent study on time-series prediction that integrates state-of-the-art techniques is:
>
> **Moreno-Pino, F., Arroyo, Á., Waldon, H., Dong, X., & Cartea, Á. (2024). Rough Transformers: Lightweight Continuous-Time Sequence Modelling with Path Signatures. _arXiv preprint_ arXiv:2405.20799.**
>
> This paper introduces the Rough Transformer, a variation of the Transformer model that operates on continuous-time representations of input sequences while achieving significantly lower computational costs. However, this model does not employ a discrete latent space suitable for enabling downstream tasks, as presented in our work.
>
> Another relevant study cited in our updated manuscript is:
>
> **Fortuin, V., Hüser, M., Locatello, F., Strathmann, H., & Rätsch, G. (2018). Som-VAE: Interpretable Discrete Representation Learning on Time Series. _arXiv preprint_ arXiv:1806.02199.**
>
> In this work, the authors address the non-differentiability challenge of discrete latent spaces similarly to VQ-VAE and introduce a Markov model in the representation space to enhance interpretability. Their evaluation spans static (Fashion-)MNIST data, time-series data of linearly interpolated (Fashion-)MNIST images, a chaotic Lorenz attractor system with two macro-states, and a medical time-series application on the eICU dataset. The eICU dataset, however, is fundamentally different from ours—it comprises lab measurements of patients aggregated from multiple critical care units across the continental United States. None of the datasets used in these studies are collected from wearable or mobile devices, nor do they feature the high heterogeneity and structured patterns of missing data that characterize the challenges addressed in our work.
>
> -   **Novelty of the work**
>
> We would like to clarify that our paper is not intended as a “method paper.” We are not proposing a novel architecture; instead, we identify VQ-VAE as an effective choice among state-of-the-art models for data imputation in heterogeneous real-world datasets with substantial missing data.
>
> Additionally, we emphasize the unique advantages of VQ-VAE for our specific application. Its discrete latent space enables downstream tasks without any fine-tuning such as suicide detection and sentiment analysis—an extension for which we already have promising results, to be included in the revised submission, and shown in the figures linked to the previous comment. Our work demonstrates that within the existing state-of-the-art, certain architectures, like VQ-VAE, are particularly well-suited for addressing the challenges posed by highly complex health time-series data recorded from wearable and mobile devices. This includes managing significant missingness and heterogeneity while facilitating solutions for diverse downstream tasks without requiring explicit training on specific patient subpopulations.
>
> The defining feature of VQ-VAE is its discrete latent space, which enables competitive performance in the two downstream tasks we evaluated. By contrast, the HMM-Transformer we included for comparison performs well in downstream prediction tasks such as sentiment analysis, for which it was designed, but struggles with suicide detection. This limitation stems from its reliance on a high-dimensional continuous latent space, where statistical changes in behavioral patterns are obscured. The complex, continuous input it provides to the CPD algorithm makes it less effective for detecting subtle shifts necessary for this application.

---

> > ### Author Response · Authors · 2024-11-22
> > **Answer to Reviewer Sgoo (2/2)**
> >
> > - **For imputation, the model should be compared to baselines. Other baselines for change-point detection than HetMM would be beneficial, e.g. point process frameworks, time series foundation models.**
> >
> > We thank the reviewer for highlighting the importance of comparisons with other state-of-the-art methods for time-series prediction and reconstruction. We acknowledge that our initial submission lacked this critical analysis, and we have addressed this by including results using the method proposed in the paper: _Arbaizar LP et al., “Emotion Forecasting: A Transformer-Based Approach,”_ JMIR Preprints, DOI: 10.2196/preprints.63962, [available here](https://preprints.jmir.org/preprint/63962). In this work, the authors used an imputation method based on hidden Markov models (HMM) for handling missing data and a Transformer deep neural network for time-series forecasting over the same database. Further details on the architecture can be found in their paper.
> >
> > To compare performance, we **applied the HMM-Transformer method by Arbaizar et al. to the suicide detection downstream tasks** presented in our study. With the VQ-VAE, we leveraged its discrete and sparse latent representations, enabling us to project data into either a single integer (indicating the closest profile) or a vector of pseudo-probabilities of a fixed length for the most relevant profiles. In contrast, the HMM-Transformer model produces a real-valued latent space with no inherent interpretability, employing embeddings of 64 dimensions with values ranging from −1 to 1. The CPD algorithm can deal with all three input types (categorical, probabilistic or real values). However, when processing the high-dimensional embeddings produced by the HMM-Transformer, the run length collapsed due to very low predictive posterior probabilities (current observation is always considered to be distinct from past data and therefore changes are prompted at all points). This behavior occurs due to the inability of the CPD to process high dimensional variables, and it evidences the need for latent representations that are low-dimensional and easy to manage, such as those returned by the VQ-VAE.
> >
> > In addition, we expanded our analysis by introducing a **new downstream task**: **sentiment analysis**, which was also performed in the paper by Arbaizar et al. In this new experiment, we excluded 758 subjects from our database during the VQ-VAE training phase. These individuals, during their respective cohort studies, logged a total of 68,219 emotions alongside daily habits recorded via passive digital phenotyping. Emotion valence—classified as 0 (negative), 1 (neutral), or 2 (positive)—was set as the target variable, predicted using all other passive variables. Predictions were made using an XGBoost classifier.
> >
> > The results of this task revealed interesting insights (see https://figshare.com/s/402cab5336581f6a6fb9. Using the same training and test partitions, the Transformer-based approach achieved a remarkable AUC score of 0.98, while the VQ-VAE also performed well, achieving an AUC of 0.81. This result is noteworthy considering that the VQ-VAE model was not explicitly designed or fine-tuned for this task, making it applicable to new patients without additional adjustments. In contrast, Arbaizar et al.’s architecture required three separate stages—each requiring training—to impute missing data (HMM), forecast the next sequence (Transformer), and classify emotions (XGBoost).
> >
> > The VQ-VAE offers a distinct advantage by combining the first two steps into a unified approach. It can project sequences with missing data directly into latent embeddings without additional intervention for imputation or preparation, demonstrating its versatility and efficiency compared to task-specific architectures.
> >
> > ### **Questions**
> >
> > - **What is the calibration of the pseudo-probabilities as compared to HetMM?**
> >
> > We didn’t quite understand what the reviewer was referring to with this question, so we were wondering if it is possible to elaborate more on it to be able to provide a precise answer. What metrics were you thinking of?
> >
> > - **How does the model deal with censoring due to loss of follow-up?**
> >
> > It is important to emphasize that our self-supervised training is conducted through autoencoder reconstruction, with the goal of reconstructing both observed and missing data. In this task, there is no censoring of data points, as we do not focus on the occurrence of any specific event. This process produces a foundation model that is subsequently employed for downstream tasks.
> >
> > As a result, our model is not specifically designed for suicide detection. Instead, we utilize the latent space representation of the patients’ heterogeneous recorded data to run a CPD algorithm, which we link to suicidal events in an unsupervised manner. Importantly, the CPD algorithm operates without direct knowledge of the suicidal events. Notably, this entire process is achieved without any fine-tuning of the VQ-VAE model.

---

> > > ### Comment · Reviewer_Sgoo · 2024-11-23
> > > **Response**
> > >
> > > Thank you for comments.
> > >
> > > As for the question: The Bayesian CPD generates probabilities for the run length, it would be helpful to see the calibration of these probabilities w.r.t. the real events.
> > >
> > > I appreciate that the authors include a related work section as well as added another baseline. I agree with the authors that with proper evaluation and benchmarking this is an interesting application of the VQ-VAE. However I maintain that as an applied use of an existing method to a specific domain and the limited technical novelty that comes with it, in my opinion, there are more appropriate venues than this conference. I will maintain my score.

---

> > > > ### Author Response · Authors · 2024-11-26
> > > >
> > > > Thank you for your prompt response.
> > > >
> > > > We understand and respect your position regarding the limited technical contribution of the method. However, we would like to point out that our submission was made to the "Applications to Neuroscience & Cognitive Science" area, where the focus, as we understand, is more on the application of methods rather than on methodological innovations. While we fully appreciate your perspective, we wanted to raise this clarification in the hope that it might offer an alternative lens through which to view our paper, given the scope of this ICLR 2025 area.

---

### Author Response · Authors · 2024-11-22
**General comments on baseline comparison, new downstream task, interpretability and reproducibility**

# GENERAL
 To address the primary concerns raised by the reviewers, we have made the following updates:

1.  **Comparison with Baselines**: We compared our approach with the HMM-Transformer model proposed by Arbaizar et al. ([DOI: 10.2196/preprints.63962](https://preprints.jmir.org/preprint/63962)). Although the HMM-Transformer model showed good performance in downstream tasks like sentiment analysis, it struggled with suicide detection due to its high-dimensional continuous latent space. In contrast, our VQ-VAE model's discrete latent space proved more interpretable and manageable, achieving competitive results across multiple tasks.

2.  **New Downstream Task**: We introduced a new task—sentiment analysis—mirroring the work by Arbaizar et al. Our VQ-VAE achieved a notable AUC of 0.81, compared to the HMM-Transformer’s 0.98, despite the VQ-VAE not being explicitly tuned for this task. This highlights its versatility in handling missing data and adaptability to different contexts without additional training steps.

3.  **Interpretability**: To address concerns about interpretability, we applied Latent Dirichlet Allocation (LDA) to the discrete latent representations learned by VQ-VAE. This analysis revealed meaningful groupings of patients based on behavioral patterns, offering insights into different types of patient behavior and variability. Details of this analysis, including figures, can be found at [https://figshare.com/s/3b97194103b7b5b6254c](https://figshare.com/s/3b97194103b7b5b6254c).

4.  **Reproducibility**: We acknowledge the challenges posed by working with private datasets. If the paper is accepted, we will release a partial dataset to enhance reproducibility.


We believe these revisions address the key concerns and elevate our work to meet ICLR's standards.

---

### Meta-Review · Area_Chair_pDmd · 2024-12-21

**Metareview:**

The paper presents a new foundation model for wearable data. The authors showcase the effectiveness of the method in modeling missing data patterns and reconstructing heterogeneous multi source time-series data. The model is used for downstream tasks on a test cohort of suicidal patients (identifying suicide attempts via a change-point algorithm).

The importance and difficulty of the clinical problem was appreciated by all the reviewers.

Concerns were rained concerning the positioning of this paper in the broader literature, and missing comparisons to other foundation models, as well as baselines that perform imputation and change point detection.

Concerning the novelty, the authors have identified it as: “no previous studies have used this type of discrete latent-space model for reconstructing heterogeneous missing data in time series from wearable and mobile devices. This unique approach is specifically motivated by the characteristics of the dataset and the downstream tasks addressed in this work.” This might be sufficient, but more arguments need to be presented concerning why existing methods do not succeed at this.

The authors have also committed to including a related work section on time series reconstruction and imputation. However, despite of the references and explanations they provided, it’s difficult to say whether this will be cogent enough to meet ICLR standards.

The sentiment analysis results are very far from the state of the art model (0.81 vs 0.98) - with the caveat that the proposed foundation model was not fine tuned for sentiment analysis. I don’t see this as evidence in favor of the proposed model being generalizable, but it’s also not evidence against it, as it’s a completely different domain. While foundation models should by definition be more generalizable than models train for a single dataset/task, there are limits to this. Maybe applying the model to some datasets and task that is closer to what it was trained on would be more useful in showing versatility (for instance: using wearables for stress detection or clinical outcome forecasting).

The authors have argued this is an application-driven paper, which, in my opinion, could be in scope for ICLR. Here is what I would recommend that the authors include for such a paper:

1. A clear exposition of the importance of the applications, its settings, and the characteristics that make it challenging; this appears to have been done well in the paper.

2. An overview of existing techniques that in the same area as the task described. Some of these techniques might not be applicable to the problem without extensive modifications, in which case clear arguments should be presented with respect to why this can’t be done. This was not done in the paper, though the authors did make a good start of it in the rebuttal.

3. For the existing methods that are applicable to (or easily modifiable for) the task, experiments showing that they fall short of solving this particular problem, motivating the methodological gap. This part was missing from the original paper. The addition of the HMM-Transformer experiment is a good step in the direction, though more methods need to be considered, including as one reviewer suggested, methods from the Gaussian Process literature).

4. Description of the new method and experiments showing that it overcomes the pitfalls of the other techniques. In this respect, the new method is well described, but without extensive comparisons, it’s difficult to judge its merits.


Overall, this paper deals with an incredibly important application, and would have potential, however the comparisons to the related work need to be improved considerably to meet the standards of ICLR. However, I encourage the authors to revise and submit to ICML.

Should the paper not be accepted at methods-focused conferences, the authors might consider sending it to venues such as MLHC, CHIL, or the ML4H Symposium at NeurIPS.

**Additional Comments On Reviewer Discussion:**

A summary of the discussion was included in the meta-review. The reviewers read the author responses, but two of them remained unconvinced about the merits of the paper.

---

### Decision · Program_Chairs · 2025-01-22

Reject